# TREM-1 links dyslipidemia to inflammation and lipid deposition in atherosclerosis

Daniel Zysset[1], Benjamin Weber[1], Silvia Rihs[1], Jennifer Brasseit[1], Stefan Freigang[1], Carsten Riether[2,3], Yara Banz[4], Adelheid Cerwenka[5], Cedric Simillion[2,6], Pedro Marques-Vidal[7], Adrian F. Ochsenbein[2,3], Leslie Saurer[1] & Christoph Mueller[1]

Triggering receptor expressed on myeloid cells-1 (TREM-1) is a potent amplifier of pro-inflammatory innate immune responses, but its significance in non-infectious diseases remains unclear. Here, we demonstrate that TREM-1 promotes cardiovascular disease by exacerbating atherosclerosis. TREM-1 is expressed in advanced human atheromas and is highly upregulated under dyslipidemic conditions on circulating and on lesion-infiltrating myeloid cells in the $Apoe^{-/-}$ mouse model. TREM-1 strongly contributes to high-fat, high-cholesterol diet (HFCD)-induced monocytosis and synergizes with HFCD serum-derived factors to promote pro-inflammatory cytokine responses and foam cell formation of human monocyte/macrophages. $Trem1^{-/-}Apoe^{-/-}$ mice exhibit substantially attenuated diet-induced atherogenesis. In particular, our results identify skewed monocyte differentiation and enhanced lipid accumulation as novel mechanisms through which TREM-1 can promote atherosclerosis. Collectively, our findings illustrate that dyslipidemia induces TREM-1 surface expression on myeloid cells and subsequently synergizes with TREM-1 to enhance monopoiesis, pro-atherogenic cytokine production and foam cell formation.

[1] Division of Experimental Pathology, Institute of Pathology, University of Bern, CH-3008 Bern, Switzerland. [2] Department of Clinical Research, Tumor Immunology, University of Bern, CH-3008 Bern, Switzerland. [3] Department of Medical Oncology, Inselspital, University Hospital and University of Bern, CH-3008 Bern, Switzerland. [4] Division of Clinical Pathology, Institute of Pathology, University of Bern, CH-3008 Bern, Switzerland. [5] Innate Immunity Group, German Cancer Research Center, D-69120 Heidelberg, Germany. [6] Interfaculty Bioinformatics Unit and SIB Swiss Institute of Bioinformatics, University of Bern, CH-3012 Bern, Switzerland. [7] Department of Internal Medicine, Internal Medicine, Lausanne University Hospital, CH-1011 Lausanne, Switzerland. Correspondence and requests for materials should be addressed to L.S. (email: leslie.saurer@pathology.unibe.ch) or to C.M. (email: christoph.mueller@pathology.unibe.ch).

Cardiovascular disease represents the leading cause of mortality in industrialized nations. The main underlying pathological process is atherosclerosis, which is increasingly recognized as a complex chronic inflammatory disease of the arteries that is critically driven by cells and mediators of the innate immune system[1]. Murine models of atherosclerosis have firmly established the causal link between hypercholesterolemia, monocytosis, lesional macrophages and atherosclerotic plaque severity[2,3]. Accordingly, a hypercholesterolemic environment directly acts on bone marrow (BM) haematopoietic stem and progenitor cells to trigger myelopoiesis[4]. Circulating monocytes are recruited to arterial vessels where the endothelium has been activated by the retention and oxidation of low-density lipoproteins (LDL)[5,6]. Following their transmigration to the intima, monocytes differentiate to macrophages and upregulate expression of scavenger receptors. Through the ingestion of oxidized LDL (oxLDL) and ensuing cholesterol accumulation, these macrophages turn into pro-inflammatory lipid-laden foam cells which critically contribute to lesion growth[6,7]. While it is accepted that mainly the pro-inflammatory Ly6C$^{hi}$ monocytes give rise to foam cells[2,3,8,9], the reported presence of distinct lesional macrophage subsets with discrete functional profiles remains difficult to apprehend[10,11], in particular, as arterial macrophages are of diverse developmental origins[12] and local proliferation also appears to contribute to the lesional macrophage pool[13].

The significance of innate immune activation in atherosclerosis and the largely non-infectious aetiology of the disease raise intriguing questions regarding the factors and receptors that can mediate the stimulation of primary monocytes, promote their differentiation into macrophages and sustain the chronic activation of lesional macrophages. Mice lacking Toll-like receptors 4 (TLR4) or myeloid differentiation primary response gene 88 (MyD88) exhibit substantially reduced plaque size[14,15], however, raising the atherosclerosis-susceptible $Apoe^{-/-}$ strain on a germ-free background does not confer protection[16]. An increasing wealth of data indeed suggests that endogenous factors representing danger-associated molecular patterns (DAMP) may play a fundamental role in the triggering of TLR, nucleotide-binding and oligomerization domain (NOD)-like receptor (NLR) family members and other pattern recognition receptors (PRR) in the context of the sterile inflammatory process in atherosclerosis[1]. Heat-shock protein 60, saturated fatty acids and modified LDL have been documented as ligands for TLR4 (refs 1,17). Experimental blockade of the DAMP molecule high-mobility group box protein-1 (HMGB1), which is released by necrotic cells and activated macrophages and is overexpressed in atherosclerotic lesions[18], reduces development of atherosclerosis[19]. Recent studies have further identified cholesterol crystal-mediated activation of the NOD-like receptor protein 3 (NLRP3) inflammasome[20,21] and fatty acid-induced mitochondrial uncoupling[22] as novel links between metabolic stress, IL-1-driven innate inflammation and atherogenesis. Importantly, emerging evidence strongly suggests that PRR act together or with other cell surface receptors to initiate and amplify signalling in atherosclerosis as illustrated by the non-redundant role of the oxLDL-binding type B scavenger receptor CD36 in coordinating NLRP3 activation and TLR4/6 assembly[23,24].

Triggering receptor expressed on myeloid cells-1 (TREM-1) potently amplifies oxidative burst and pro-inflammatory cytokine secretion when signalling in concert with other PRR[25,26]. The role of TREM-1 has mostly been acknowledged from acute infection models where blockade of TREM-1 conferred significant protection[27]. However, data generated by us and others strongly suggest—in line with the identification of HMGB1 and multimerized peptidoglycan recognition protein-1 (PGLYRP1) as potential non-bacterial ligands[28–30]—that TREM-1 may also substantially contribute to chronic and non-infectious inflammatory conditions[29,31–40]. On the basis of the prominent expression of TREM-1 by neutrophils and monocytes and its synergistic action with both TLRs and PRRs of the NACHT-LRR class[25,26], we hypothesized that TREM-1 represents a central player for innate immune activation in atherosclerosis. Here, we demonstrate that TREM-1 indeed aggravates diet-induced atherogenesis. Mechanistically, TREM-1-mediated signalling exacerbates monocytosis in vivo by skewing myeloid differentiation towards increased monocyte output. Moreover, TREM-1 stimulation potently impacts on the expression of CD36 as well as other genes involved in lipid metabolism and augments the pro-inflammatory cytokine responses and foam cell formation of human monocytes/macrophages in vitro. Our findings establish a distinct role for TREM-1 in a chronic inflammatory disorder of non-infectious origin and reveal two so far unappreciated effects of TREM-1 on myelopoiesis and cellular cholesterol metabolism.

## Results

**TREM-1 aggravates atherosclerosis.** We assessed the potential role of TREM-1 in atherogenesis using the $Trem1^{-/-}$ mouse strain that was recently generated and characterized by our laboratory[31]. $Trem1^{-/-}$ mice were backcrossed onto the atherosclerosis-susceptible $Apoe^{-/-}$ background and 6–8-week-old female $Trem1^{-/-}$ $Apoe^{-/-}$ and $Trem1^{+/+}$ $Apoe^{-/-}$ mice were concurrently placed on a high-fat, high-cholesterol-containing diet (HFCD). At 16 weeks post HFCD feeding, $Trem1^{-/-}$ $Apoe^{-/-}$ mice exhibited a 40% reduction in the overall extent of atherosclerosis in the aorta (Fig. 1a,b). The decreased atherosclerotic surface area in $Trem1^{-/-}$ $Apoe^{-/-}$ mice was due to a substantially smaller average lesion size rather than being associated with reduced numbers of individually enumerable lesions (Fig. 1c,d). While the effect of the TREM-1 deficiency was clearly apparent throughout the aorta, this difference was not reflected by the cross-sectional lesion area at the aortic root (Fig. 1e,f). Atherosclerotic lesions usually develop first at this anatomical site and had, at 16 weeks post HFCD, already progressed to very advanced plaques, thus very likely masking the effect of TREM-1 deficiency on atherogenesis.

The pro-atherogenic impact of TREM-1 was not mediated by effects on lipid metabolism, since HFCD-fed $Trem1^{-/-}$ $Apoe^{-/-}$ and $Trem1^{+/+}$ $Apoe^{-/-}$ mice exhibited similar total cholesterol, HDL-cholesterol and LDL-cholesterol levels (Fig. 1g). Moreover, HFCD-fed $Trem1^{-/-}$ $Apoe^{-/-}$ and $Trem1^{+/+}$ $Apoe^{-/-}$ mice showed comparable fasting blood glucose levels and similar glucose clearance upon oral glucose challenge, which indicated equal insulin sensitivity in both groups of mice (Supplementary Fig. 1e–g). Taken together, these findings suggested a major impact of TREM-1 on atherosclerotic lesion progression but ruled out systemic effects on lipid or glucose metabolism as an underlying mechanism.

**TREM-1 exacerbates dyslipidemia dependent monocytosis.** In experimental models of diet-induced atherogenesis the severity of atherosclerosis closely correlates with increased numbers of circulating monocytes and neutrophils[2,41]. We therefore analysed the absolute numbers of peripheral blood myeloid cell subsets after 4 and 16 weeks of HFCD feeding (Fig. 2a and Supplementary Fig. 2a). HFCD-feeding induced a time-dependent increase in circulating Ly6C$^{hi}$ monocytes in both groups of mice (Fig. 2a). However, this HFCD-induced monocytosis was drastically reduced in mice lacking TREM-1

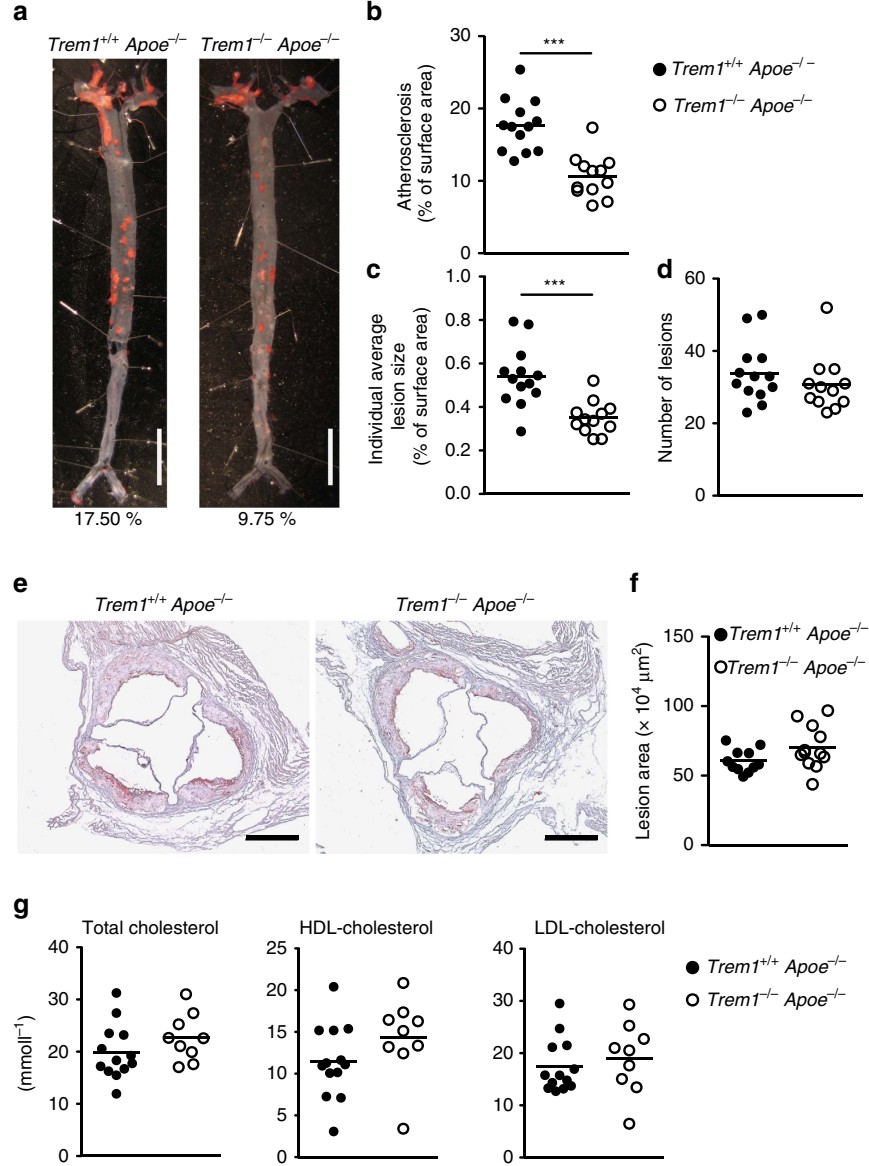

**Figure 1 | TREM-1 promotes atherogenesis in absence of an altered lipid metabolism. (a–d)** Extent of atherosclerosis in female $Trem1^{+/+}$ $Apoe^{-/-}$ ($n = 13$) and $Trem1^{-/-}$ $Apoe^{-/-}$ mice ($n = 12$) at 16 weeks post HFCD feeding as assessed by computational analysis of Oil-red O (ORO) stained aortas. **(a)** Selected examples of *en face* preparations of the aortas; for each group of mice, an aorta with a representative extent of atherosclerosis is shown. Scale bars, 10 mm. **(b)** Overall extent of atherosclerosis (aortic lesion surface area expressed as % of total aortic surface). **(c)** Mean individual lesion size per individual mouse and aorta. **(d)** Number of enumerable lesions per aorta. **(b–d)** Circles show individual data for mice from three pooled independent experiments, lines indicate mean values per group of mice. **(e,f)** Extent of atherosclerotic lesions in the aortic root at 16 weeks post HFCD feeding. **(e)** Representative examples of ORO-stained sections of the aortic root. Scale bars, 500 μm. **(f)** The lesion area was calculated from 10 sequential ORO-stained sections for each $Trem1^{+/+}$ $Apoe^{-/-}$ ($n = 11$) and $Trem1^{-/-}$ $Apoe^{-/-}$ ($n = 12$) mouse. Circles show data for individual mice. **(g)** Fasting serum concentrations of total cholesterol, HDL-cholesterol and LDL cholesterol of $Trem1^{+/+}$ $Apoe^{-/-}$ ($n = 13$) and $Trem1^{-/-}$ $Apoe^{-/-}$ ($n = 12$) mice at 16 weeks post HFCD feeding. \*\*\*$P < 0.001$ as determined by the two-tailed $t$-test. Statistically not significant differences with $P > 0.05$ are not indicated.

(Fig. 2a). HFCD feeding also resulted in a marked neutrophilia that, conversely, appeared to be more pronounced in $Trem1^{-/-}$ $Apoe^{-/-}$ mice (Fig. 2a). To investigate the underlying mechanisms behind the aggravated monocytosis in $Trem1^{+/+}$ $Apoe^{-/-}$ mice, we examined the BM of 16-week chow diet or HFCD-fed mice (Fig. 2b–d and Supplementary Fig. 2b). Under chow-diet feeding, common myeloid progenitors (CMP) were phenotypically increased while granulocyte/macrophage progenitors (GMP) were significantly reduced in frequencies in $Trem1^{-/-}$ $Apoe^{-/-}$ mice (Fig. 2b). In contrast, frequencies of both CMPs and GMPs were decreased in $Trem1^{-/-}$ $Apoe^{-/-}$ mice under HFCD (Fig. 2b). Frequencies of lineage-negative

(lin$^-$) cKit$^{hi}$ Sca1$^+$ stem/progenitor cells (LSK) were comparable between the two groups independently of the diet (Fig. 2b). To determine whether the enforced monocytosis in $Trem1^{+/+}$ $Apoe^{-/-}$ mice was related to functional alterations in haematopoietic stem/progenitor cells, we determined myeloid BM colony-forming unit (CFU) capacity under chow diet and HFCD feeding. Both groups of mice had comparable CFUs irrespective of the diet (Fig. 2c). Surprisingly, however, HFCD-fed $Trem1^{+/+}$ $Apoe^{-/-}$ mice formed more monocyte colonies (M) at the expense of granulocyte colonies (G) (Fig. 2d,e). In line with this finding, GMP from HFCD-fed $Trem1^{+/+}$ $Apoe^{-/-}$ mice exhibited increased mRNA for *Irf8*, which is a key regulator of

monocytic over granulocytic lineage differentiation (Fig. 2f)[42]. GMP, but not CMP or LSK cells, also expressed distinct levels of surface TREM-1 (Supplementary Fig. 2c)[31]. TREM-1 expression on GMP was not further increased by HFCD compared with chow diet feeding (Supplementary Fig. 2c); moreover, activation of sorted GMP by plate-bound anti-TREM-1 in presence or absence of HFCD serum did not augment monocyte differentiation *in vitro* (Supplementary Fig. 2d,e). These results indicated that the effect of TREM-1 on skewed monocyte differentiation was potentially indirect and/or upstream of GMP.

Considering the synergistic impact of dyslipidemia and TREM-1 on the peripheral immune compartment, we in turn

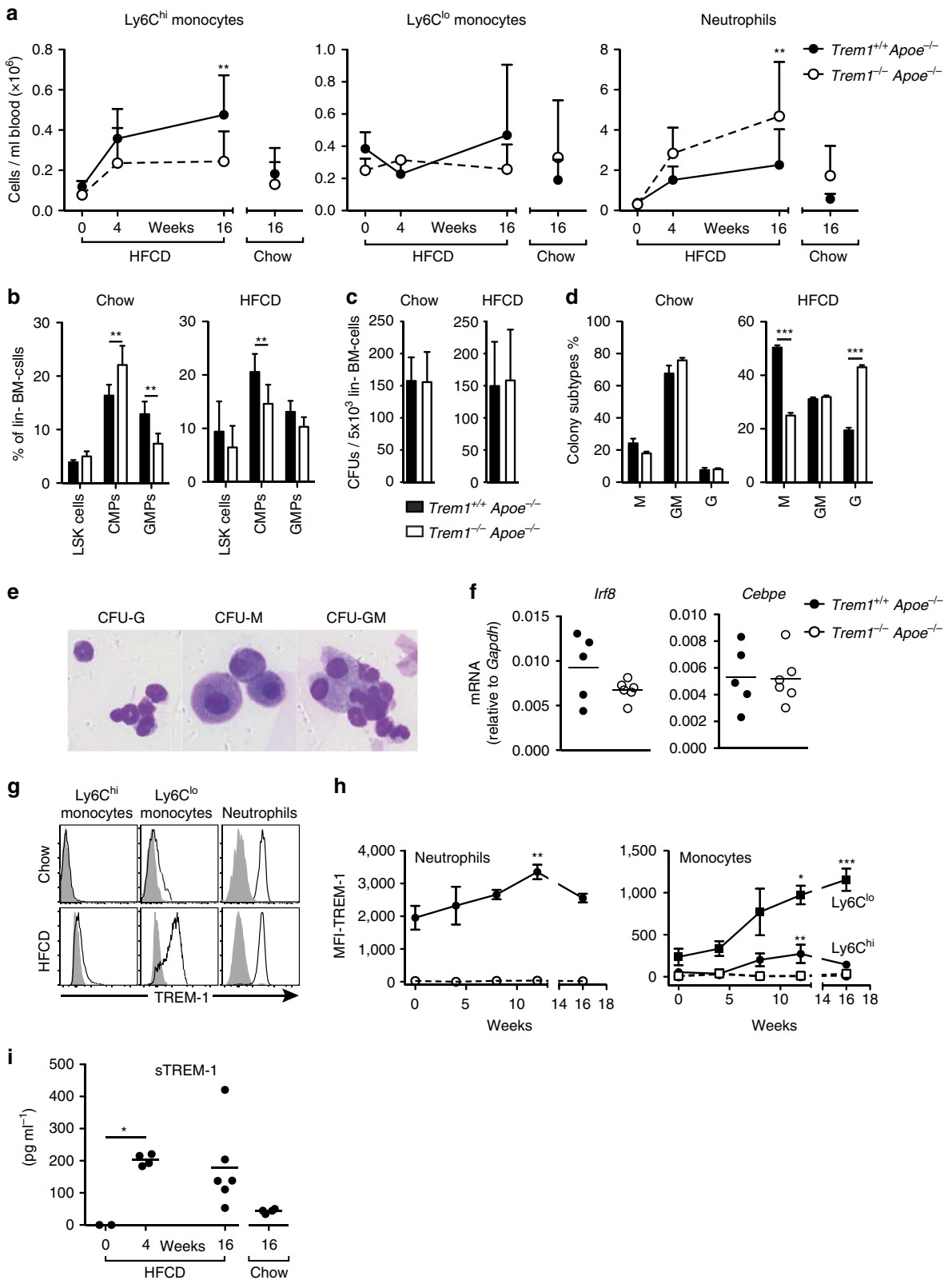

assessed the potential impact of HFCD feeding on TREM-1 surface expression by peripheral blood myeloid cell subsets. Concordant with previous findings[31,43], TREM-1 was highly expressed on neutrophils and to a lower degree on Ly6C$^{lo}$ monocytes in chow-fed mice while no surface expression was detected on Ly6C$^{hi}$ monocytes (Fig. 2g). Intriguingly, HFCD feeding markedly upregulated TREM-1 expression on all myeloid subsets. The increase was most prominent for neutrophils and Ly6C$^{lo}$ monocytes, but was also evident for Ly6C$^{hi}$ monocytes (Fig. 2g,h). The modulated expression of surface TREM-1 was paralleled by an increase in serum sTREM-1 that was apparent already at 4 weeks post HFCD (Fig. 2i).

Taken together, these results suggested that dyslipidemia distinctly increases expression of TREM-1 on all circulating myeloid cell subsets and that TREM-1-mediated signalling in turn provokes skewed monocyte differentiation in the BM, resulting in exacerbated monocytosis and atherogenesis.

**TREM-1 augments aortic macrophage accumulation.** While our data so far suggested a contribution of TREM-1 to the HFCD-induced monocytosis, we also sought to investigate a potential contribution of TREM-1-expressing cells within the progressing atherosclerotic plaque. *Trem1* mRNA was barely detectable in the aortic arches of young (before HFCD) or of 16-week chow diet-fed *Trem1*$^{+/+}$ *Apoe*$^{-/-}$ (Fig. 3a). In contrast, 16-week HFCD feeding resulted in a markedly upregulated expression of TREM-1 mRNA (Fig. 3a and Supplementary Fig. 3a). The increased abundance of *Trem1* transcripts was paralleled by a substantial appearance of *Cd68* but not *Ly6g* expression (Fig. 3a), illustrating that lesion formation and vascular monocyte-macrophage accumulation rather than neutrophil infiltration correlated with *Trem1* expression. In line with this, also expression of *Trem2* but not *Treml1* or *Treml4* was higher in the aortic arch of 16-week HFCD-fed *Trem1*$^{+/+}$ *Apoe*$^{-/-}$ mice compared to *Trem1*$^{-/-}$ *Apoe*$^{-/-}$ mice (Supplementary Fig. 3b).

The parallel increase in *Trem1* and *Cd68* transcripts at 16-weeks post HFCD indicated that not only the TREM-1-driven peripheral monocytosis but also local TREM-1-mediated activation of monocyte/macrophages could contribute to atherosclerotic lesion development. To further elucidate potential mechanisms underlying TREM-1-stimulated lesion progression, we subjected aortic tissues to a Nanostring-based gene expression profiling, using the nCounter Mouse Immunology panel supplemented with 30 genes of interest (Supplementary Table 1). Since the extremely low number of macrophages that were retrieved from digested aortas precluded gene expression analysis in isolated cells, we directed our analysis to the aortic

arch where the least differences in total lesion size were observed between the two groups of mice at 16-weeks post HFCD feeding. Principal component analysis of the NanoString data indicated that the largest possible variance was induced by HFCD compared to chow diet feeding (Fig. 3b). However, among HFCD-fed mice, variances in gene expression were nonetheless evident between the *Trem1*$^{+/+}$ *Apoe*$^{-/-}$ or *Trem1*$^{-/-}$ *Apoe*$^{-/-}$ group (Fig. 3b,c and Supplementary Table 2). As anticipated from Fig. 2a, the aggravated monocytosis in *Trem1*$^{+/+}$ *Apoe*$^{-/-}$ mice primarily translated into increased aortic expression of multiple monocyte/macrophage-related markers (Fig. 3c and Supplementary Fig. 3c). Accordingly, also the expression of several pro-inflammatory cytokines, oxidases and genes involved in foam cell formation was augmented (Fig. 3c and Supplementary Fig. 3c). Normalization of these differentially expressed genes to monocyte/macrophage-related markers did not demonstrate significant differences between the two groups of mice (Supplementary Fig. 4). Hence, the distinct transcriptional profile of whole aortic tissue from HFCD-fed *Trem1*$^{+/+}$ *Apoe*$^{-/-}$ mice was mainly reflective of increased monocyte/macrophage accumulation rather than altered gene expression patterns on a per cell level. While these results did not indicate an unequivocal mechanism for a local contribution of TREM-1 to atherosclerotic lesion progression, the data firmly established increased monocyte/macrophage infiltration as a major pro-atherogenic consequence of TREM-1-mediated signalling in HFCD-fed *Trem1*$^{+/+}$ *Apoe*$^{-/-}$ mice.

**TREM-1 affects the composition of aortic myeloid cell subsets.** Enhanced expression of *Trem1* at arterial sites was related to increased monocyte/macrophage accumulation (Fig. 3a,c and Supplementary Fig. 3). However, lesional macrophages are likely composed of heterogeneous populations comprising differentiated macrophage foam cells but also recently recruited monocytes, which could differ in their expression of TREM-1. To gain insight into potential roles of TREM-1-expressing myeloid cell subsets at arterial sites, we aimed to identify the major TREM-1-expressing cell subset in atherosclerotic lesions. In addition, we sought to determine whether TREM-1 could contribute to lesion progression by influencing the composition of plaque-infiltrating monocyte/macrophage populations. We chose a flow cytometry approach to analyse enzymatically digested aortas derived from *Trem1*$^{+/+}$ *Apoe*$^{-/-}$ and *Trem1*$^{-/-}$ *Apoe*$^{-/-}$ mice at 16 weeks post HFCD feeding. Among single, live- non-autofluorescent cells we pre-gated on CD45$^{+}$ CD11b$^{+}$ cells and subsequently excluded MHCII$^{+}$ CD11c$^{+}$ cells which represented dendritic cells[12,44] (Supplementary Fig. 5a). Within the remaining CD11b$^{+}$ population we distinguished five myeloid

**Figure 2 | TREM-1 drives skewed monocyte differentiation and HFCD-dependent monocytosis.** (**a**) Absolute numbers of peripheral blood myeloid cell subsets before HFCD feeding (week 0; *n* = 4), after 4 weeks of HFCD-feeding (*n* = 4–5), after 16 weeks of HFCD-feeding (*n* = 13–15) and after 16 weeks of chow-feeding (*n* = 9–13) as determined by flow cytometry. Circles represent mean values + s.d. for each group. Data are pooled from three independent experiments for the 16-week time-point and from one experiment for the 0 and 4 week analyses. (**b**) Relative frequencies of LSK cells (Sca1$^{+}$ cKit$^{hi}$), CMP (Sca1$^{-}$ cKit$^{hi}$ FcgR$^{lo}$ CD34$^{+}$) and GMP (Sca1$^{-}$ cKit$^{hi}$ FcgR$^{+}$ CD34$^{+}$) cells among lineage$^{-}$ (lin$^{-}$; Ter119$^{-}$, CD3e$^{-}$, Gr1$^{-}$ and B220$^{-}$) and CD127$^{-}$ BM cells isolated from 16-week-HFCD-fed or chow-fed mice. Bars show mean values + s.d. of *n* = 6–9 HFCD-fed mice (two independent experiments) and *n* = 4–5 chow-fed controls. (**c**) Myeloid colony-forming units per well of 5 × 10$^{3}$ plated lin$^{-}$ BM cells and (**d**) relative frequencies of colony subtypes (M: monocyte colonies; G: granulocyte colonies; GM: mixed monocyte/granulocyte colonies). Data show mean values + s.d. of *n* = 6–9 HFCD-fed mice (from two independent experiments) and from *n* = 4–5 chow-fed mice per group. (**e**) Representative examples of CFU-G, CFU-M and CFU-GM. (**f**) mRNA expression of *Irf8* and *Cebpe* in GMP isolated from 16-week-HFCD-fed mice. (**g**) Representative histograms for TREM-1 surface expression (lines) versus isotype control staining (filled) on peripheral blood neutrophils, Ly6C$^{hi}$ and Ly6C$^{lo}$ monocytes from *Trem1*$^{+/+}$ *Apoe*$^{-/-}$ mice 16 weeks post chow or HFCD feeding. (**h**) Median fluorescence intensity (MFI) of TREM-1 surface expression (with subtracted MFI values of matched isotype control-stained cells) on peripheral blood myeloid cell subsets at the indicated time points post HFCD feeding in *Trem1*$^{+/+}$ *Apoe*$^{-/-}$ (black symbols) and *Trem1*$^{-/-}$ *Apoe*$^{-/-}$ (open symbols) mice. Symbols indicate mean + s.d. of *n* = 5 mice and data are representative of three independent experiments for *Trem1*$^{+/+}$ *Apoe*$^{-/-}$ mice. (**i**) serum soluble TREM-1 was determined by ELISA. *$P$ < 0.05, **$P$ < 0.01, ***$P$ < 0.001 as determined by the two-way ANOVA test (two-tailed *t*-test for **f**). Statistically not significant differences with $P$ > 0.05 are not indicated.

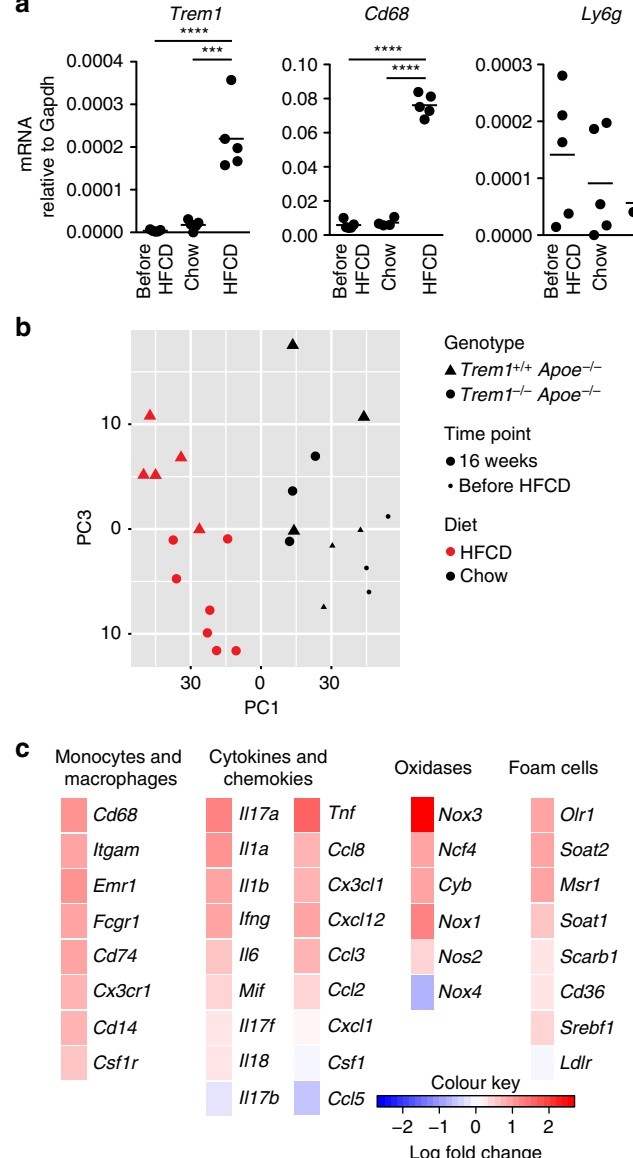

**Figure 3 | *Trem1* is expressed *in situ* and associates with increased macrophage accumulation.** (**a**) *Trem1*, *Cd68* and *Ly6g* mRNA expression in the aortic arch of *Trem1*$^{+/+}$ *Apoe*$^{-/-}$ mice before the start of diet (before HFCD) and at 16 weeks post chow or HFCD feeding was determined by qRT-PCR and normalized to the expression of *Gapdh*. ***$P < 0.001$, ****$P < 0.0001$. (**b,c**) RNA isolated from the aortic arch of *Trem1*$^{+/+}$ *Apoe*$^{-/-}$ and *Trem1*$^{-/-}$ *Apoe*$^{-/-}$ mice before HFCD ($n = 3$ per group) or at 16 weeks post chow ($n = 3$ per group) or post HFCD feeding ($n = 5$ and 7) was subjected to Nanostring nCounter-based quantification of gene expression. (**b**) PCA analysis. Each symbol represents an individual mouse. (**c**) Heat-maps of selected gene groups. Mean log fold changes in gene expression in the aortic arch of 16-week HFCD-fed *Trem1*$^{+/+}$ *Apoe*$^{-/-}$ ($n = 5$) over *Trem1*$^{-/-}$ *Apoe*$^{-/-}$ ($n = 7$) mice are shown. PCA, principle component analysis.

cell subsets based on their expression of Ly6C and MHC class II: 1. Neutrophils (Ly6C$^{int}$, MHCII$^-$), which were also Ly6G$^+$ (Supplementary Fig. 5b), 2. Ly6C$^{hi}$ monocytes (Ly6C$^{hi}$, MHCII$^-$) 3. Ly6C$^{int}$ macrophages (Ly6C$^{int}$ MHCII$^+$) 4. MHCII$^+$ macrophages (Ly6C$^-$ MHCII$^+$) and 5. MHCII$^-$ macrophages (Ly6C$^-$, MHCII$^-$) (Fig. 4a,b). The identity of macrophages was confirmed by staining for CD64 (or F4/80 in some experiments), which was largely absent on neutrophils and

expressed at higher levels in macrophages as compared to Ly6C$^{hi}$ monocytes (Fig. 4b and Supplementary Fig. 5c). Importantly, the segregation of myeloid subsets according to their Ly6C and MHC class II expression was adapted from the monocyte waterfall concept of macrophage differentiation[45,46], which also considers newly recruited Ly6C$^{hi}$ monocytes and Ly6C$^{int}$ macrophage differentiation stages. In contrast, a general pre-gating of F4/80$^+$ cells *a priori* excludes these subsets since Ly6C$^{hi}$ monocytes and Ly6C$^{int}$ cells unlike MHCII$^+$ and MHCII$^-$ macrophages are F4/80$^{lo/-}$ (Supplementary Fig. 5c)[12]. As expected, TREM-1 was expressed on aortic neutrophils, albeit at substantially lower levels compared with peripheral blood neutrophils (Figs 2g and 4c). This likely related to the enzymatic digestion protocol of aortas. Strikingly, however, surface TREM-1 expression was also clearly detectable on Ly6C$^{int}$ MHCII$^+$ cells as well as on Ly6C$^-$ MHCII$^-$ lesional macrophages, whereas Ly6C$^-$ MHCII$^+$ macrophages did not appear to express substantial levels of TREM-1 (Fig. 4c,d).

We next compared the relative abundance of individual myeloid cell subsets in aortic plaques of HFCD-fed *Trem1*$^{+/+}$ *Apoe*$^{-/-}$ versus *Trem1*$^{-/-}$ *Apoe*$^{-/-}$ mice. As anticipated from their reduced aortic lesion size and lower expression of macrophage-associated genes (Figs 1a–c and 3c), aortic plaques of *Trem1*$^{-/-}$ *Apoe*$^{-/-}$ mice contained significantly less CD11b$^+$ CD11c$^-$ cells (Fig. 4e). This difference appeared to be mainly due to a significant reduction in the percentage of Ly6C$^-$ MHC II$^-$ macrophages in lesions of *Trem1*$^{-/-}$ *Apoe*$^{-/-}$ mice (Fig. 4e). In contrast, deficiency in TREM-1 did not alter the relative abundance of infiltrating neutrophils and Ly6C$^{hi}$ monocytes (Fig. 4e). Taken together, these results indicated that TREM-1 is expressed by aortic wall-infiltrating neutrophils, Ly6C$^{int}$ MHCII$^+$ as well as Ly6C$^-$ MHCII$^-$ macrophages and that the cellular composition in *Trem1*$^{+/+}$ *Apoe*$^{-/-}$ versus *Trem1*$^{-/-}$ *Apoe*$^{-/-}$ mice mostly differed in the relative abundance of aortic Ly6C$^-$ MHCII$^-$ macrophages.

**TREM-1 and dyslipidemic factors interact reciprocally.** In aortic lesions of *Trem1*$^{+/+}$ *Apoe*$^{-/-}$ mice, highest MFI values for surface TREM-1 were found on neutrophils, followed by Ly6C$^{int}$ MHCII$^+$ cells (Fig. 4c,d), which we regarded as monocytes in the process of differentiating to mature macrophages based on the sequential monocyte waterfall concept of intestinal macrophage differentiation[45,46]. Indeed, in the inflamed colon highest TREM-1 expression can be detected on Ly6C$^{hi}$ effector monocytes followed by their Ly6C$^{int}$ descendants[47]. Since peripheral blood Ly6C$^{hi}$ monocytes have been established as precursors for lesional macrophages[2,3,8,9], yet in the mouse even under HFCD feeding express little surface TREM-1 (Fig. 2g), we considered the possibility that TREM-1 is upregulated upon transendothelial migration of Ly6C$^{hi}$ monocytes to the subendothelial space. The uptake of oxLDL by infiltrating monocytes represents a key factor in driving macrophage foam cell differentiation and atherogenesis[48,49]. Indeed, exposure to purified endotoxin-free oxLDL, but also to heat-inactivated dyslipidemic serum derived from HFCD-fed mice (HFCD serum), which contains minimally modified LDL circulating *in vivo*, significantly upregulated the expression of surface TREM-1 on human primary monocytes (Fig. 5a,b). Moreover, mean TREM-1 MFI values for oxLDL-stimulated cells even exceeded those observed for cells stimulated with lipopolysaccharide (LPS) (Fig. 5b). Considering that oxLDL represented a potential non-microbial factor capable of upregulating TREM-1 (ref. 50,51), we next addressed whether oxLDL and HFCD serum-derived factors could synergize with TREM-1-mediated signals to amplify production of

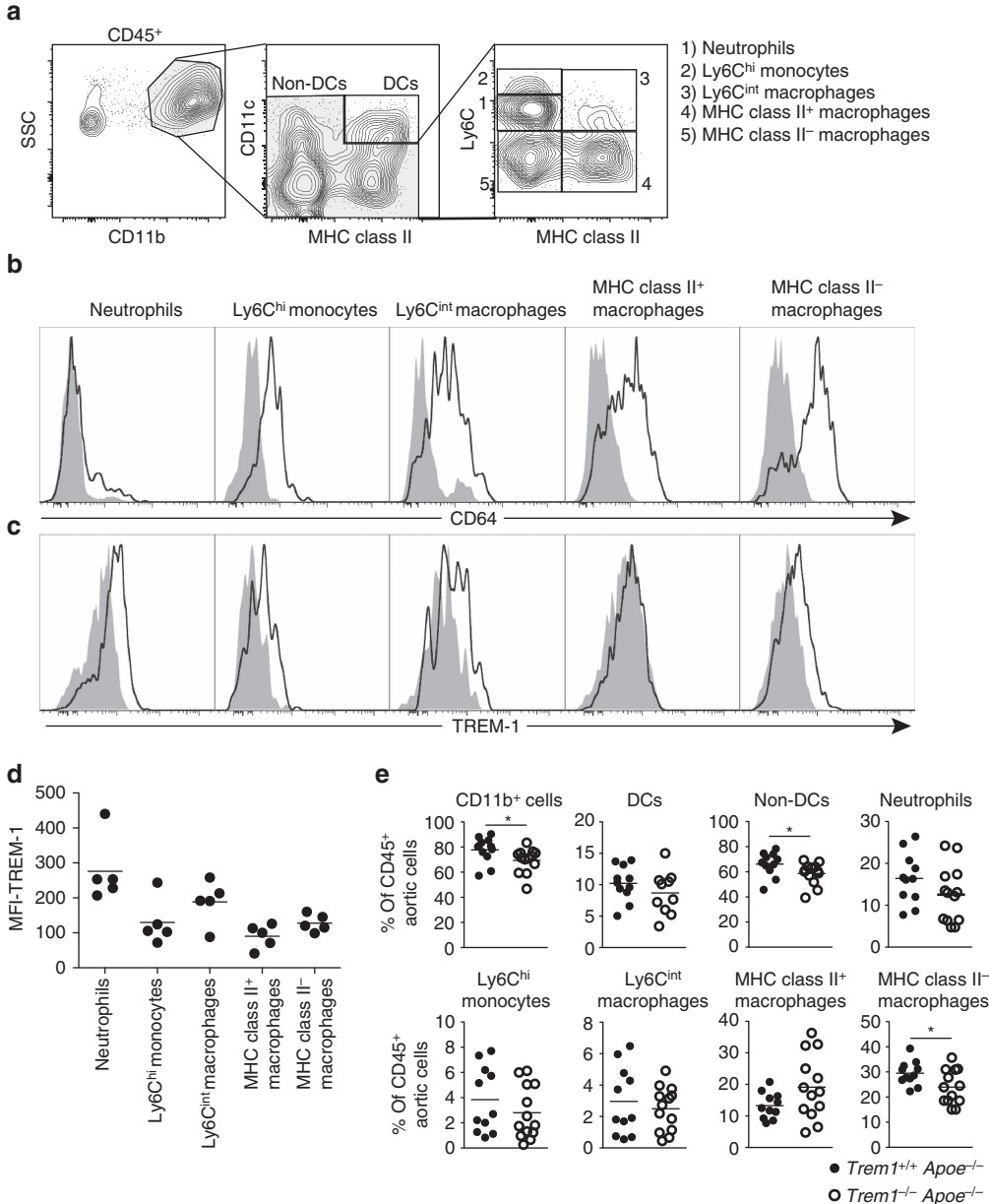

**Figure 4 | TREM-1 influences the composition of aortic myeloid cell subsets. (a–e)** Flow cytometry analysis of aortic wall-infiltrating myeloid cell subsets isolated from individually digested aortas of 16-week HFCD-fed $Trem1^{+/+}$ $Apoe^{-/-}$ and $Trem1^{-/-}$ $Apoe^{-/-}$ mice. (**a**) Representative gating strategy for identification of myeloid cell subsets within single, live, non-autofluorescent CD45$^+$, CD11b$^+$ non-dendritic cells ($Trem1^{+/+}$ $Apoe^{-/-}$). Identity of neutrophils was confirmed by inclusion of Ly6G in the staining panel. (**b**) Representative staining panels for expression of the monocyte/macrophage marker CD64 (line) on the indicated cell subsets (filled histograms: matched isotype control-stained cells) ($Trem1^{+/+}$ $Apoe^{-/-}$). (**c**) Representative staining panels for expression of surface TREM-1 (line) on the indicated cell subsets (filled histograms: matched isotype control-stained cells) ($Trem1^{+/+}$ $Apoe^{-/-}$). (**d**) MFI values for TREM-1 surface expression (with subtracted MFI values of matched isotype control-stained cells) on the indicated myeloid cell subsets of $Trem1^{+/+}$ $Apoe^{-/-}$ mice. Circles represent data for individual mice, lines indicate mean values per group. (**e**) Relative abundance (% among CD45$^+$ cells) of arterial wall-infiltrating cell subsets in 16-week HFCD-fed $Trem1^{+/+}$ $Apoe^{-/-}$ versus $Trem1^{-/-}$ $Apoe^{-/-}$ mice. Circles represent data for individual mice from three independent pooled experiments, lines indicate mean values per group. *$P < 0.05$ as determined by the two-tailed $t$-test. Statistically not significant differences with $P > 0.05$ are not indicated.

pro-atherogenic cytokines, analogous to the well-established synergistic effects of TREM-1 with LPS[27]. Incubation of monocytes with oxLDL or HFCD serum alone did not induce substantial secretion of IL-1α, IL-1β or TNFα, whereas anti-TREM-1 stimulation in the absence of other factors distinctly increased the production of TNFα but not of IL-1 cytokines (Fig. 5c). As anticipated, TREM-1 stimulation synergized with LPS for increased cytokine production. Intriguingly, exposure to the HFCD serum strongly augmented the TREM-1-induced secretion of IL-1α, IL-1β or TNFα while no such effect was observed for oxLDL stimulation (Fig. 5c). This suggested that the minimally modified LDL present in the dyslipidemic HFCD serum rather than in vitro generated, extensively oxidized LDL may substitute for LPS to synergize with TREM-1 stimulation for increased pro-inflammatory cytokine production from monocyte/macrophages.

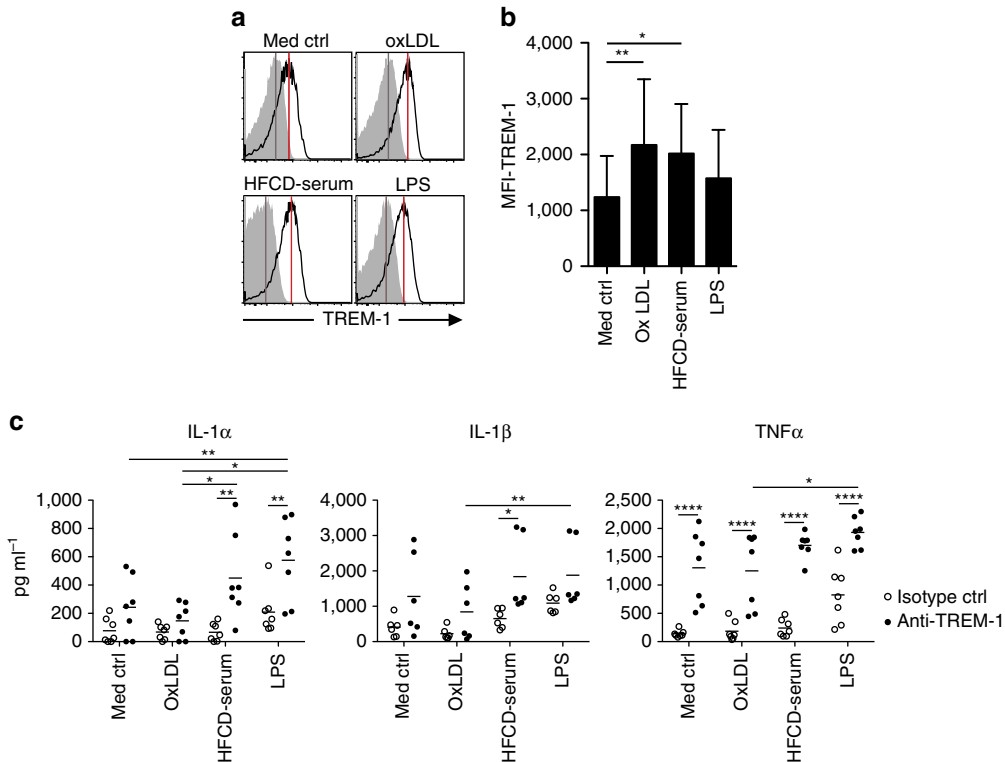

**Figure 5 | TREM-1 synergizes with dyslipidemic factors for pro-inflammatory cytokine secretion. (a–c)** Human CD14[hi] monocytes were FACS-purified from buffy coats obtained from different blood donors. (**a**) Representative histograms for TREM-1 surface expression (lines; filled histograms represent isotype control stained cells) on human peripheral blood CD14[hi] monocytes after 20 h of stimulation with the indicated stimuli. (**b**) MFI values for TREM-1 surface expression after subtraction of MFI values for matched isotype control-stained cells. Bars show mean values of seven independent experiments with error bars indicating the s.d. (**c**) Production of IL-1α, IL-1β and TNFα by CD14[hi] monocytes after 20 h of culture with the indicated factors. Symbols show individual values of 6–7 independent experiments and different blood donors. *P < 0.05, **P < 0.01, ****P < 0.0001 as determined by the one-way (**b**) or two-way (**c**) ANOVA test. Statistically not significant differences with P > 0.05 are not indicated.

**TREM-1 enhances foam cell formation *in vitro*.** IL-1 family cytokines have been ascribed a central role in atherogenesis[52], however, neither *Il1a* nor *Il1b* were among the 111 significantly TREM-1-regulated genes identified in the Nanostring analysis (Supplementary Table 2). We therefore considered additional mechanisms that could account for a local TREM-1-mediated lesion progression and evaluated the possibility that TREM-1-mediated signals directly contribute to foam cell differentiation. Because of the difficulty in obtaining sufficient TREM-1-expressing primary murine monocytes and the absence of TREM-1 on peritoneal and BM-derived murine macrophages, we used the human myelomonocytic cell line U937 that was stably transduced with TREM-1 and DAP12 (referred to as U937-TD) by Tessarz *et al.*[53]. Incubation of U937-TD in the presence of murine HFCD serum and an isotype control antibody for 48 h resulted in detectable but rather inefficient foam cell formation as assessed by Oil Red O staining (Fig. 6a,b). In contrast, stimulation of U937-TD cells with an agonistic anti-TREM-1 antibody lead to a significant increase of cells exhibiting the characteristic lipid droplets (Fig. 6a,b). The effect was specific to TREM-1-mediated signalling as incubation of U937-TD cells with LPS did not substantially enhance foam cell formation (Fig. 6a,b). Foam cell formation is primarily mediated through uptake of modified LDL through specific receptors[49]. Crosslinking of TREM-1 on U937-TD in fact potently upregulated expression of CD36 irrespective of the absence or presence of 5% HFCD serum in the medium (Fig. 6c,d). We next reverted to the use of primary human CD14[hi] monocytes as more representative precursors for foam cells. Although the foam cell

formation capacity of *in vitro* cultured primary monocytes was generally less efficient, stimulation with anti-TREM-1, but not LPS, clearly augmented the lipid droplet content (Fig. 6e,f). Strikingly, TREM-1-mediated activation of human primary monocytes not only increased mRNA expression of *CD36* but also of other receptors implicated in lipid uptake such as *MSR1* and *LDLR* (Fig. 6g). Moreover, TREM-1-activated monocytes showed reduced expression of the cholesterol efflux-related genes *ABCA1* and *ABCG1* and altered mRNA expression of several intracellular cholesterol transport proteins such as *NPC1*, *NPC2* and *STARD4* (Fig. 6g), the latter likely reflecting a compensatory response to the increased intracellular lipid content.

Together, these results identified a so far unappreciated direct and potent impact of TREM-1 on foam cell formation *in vitro* which not only related to increased expression of specific scavenger receptors but was generally associated with an altered expression of genes involved in oxLDL uptake and cellular cholesterol metabolism.

**TREM1 is expressed in established human atheromas.** Our data so far demonstrated that TREM-1 promotes atherosclerotic lesion progression in an experimental atherosclerosis model *in vivo* by aggravating monocytosis and monocyte/macrophage infiltration of aortas. However, the *in situ* expression of TREM-1 in aortic lesions as well as the potent impact of TREM-1 on foam cell generation *in vitro* suggested that TREM-1-mediated effects indeed contribute to lesion growth locally. Since in a Russian population-based cohort certain polymorphisms in the *TREM1*

gene associated with a reduced risk for coronary artery disease[54], we assessed the potential expression of TREM-1 in human atheromas. To this end, we performed quantitative real-time PCR (qRT-PCR) analyses on RNA isolated from human formalin-fixed paraffin-embedded (FFPE) aortic tissue specimen displaying either only minor alterations or severe atherosclerosis (Fig. 7a). Similar to our findings in control or chow-fed versus HFCD-fed $Trem1^{+/+} Apoe^{-/-}$ mice (Fig. 3a), $TREM1$ mRNA was barely expressed in human aortic tissues with minor histopathological alterations (Fig. 7b). In contrast, mRNA levels for TREM-1 and

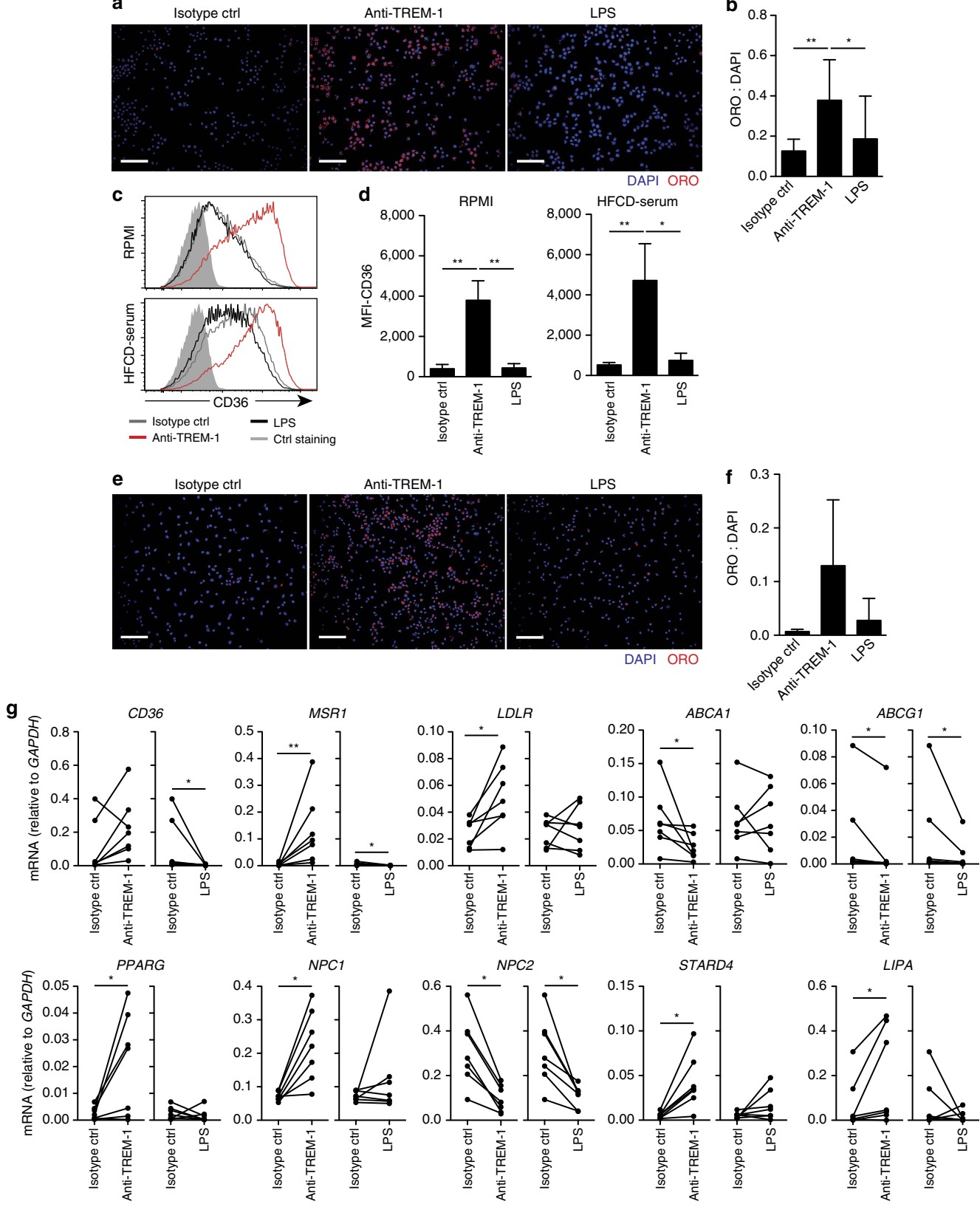

CD68 were significantly upregulated in advanced atheromas, while expression of mRNA for the granulocyte marker CD15 was not substantially increased (Fig. 7b). Hence, analogous to what we observed in murine atherosclerotic lesions, lesional macrophages appeared to represent a major population of TREM-1-expressing cells also in human atheromas.

Collectively, these results from human atherosclerosis patients and our findings from an experimental model of atherogenesis, where deficiency in TREM-1 conferred significant protection, establish a major pathogenic role for TREM-1 in atherosclerosis. We show that factors contained in dyslipidemic serum are sufficient to modulate expression of TREM-1 and that TREM-1 in turn can synergize with dyslipidemia for augmented inflammation and foam cell formation. Based on the distinct impact of TREM-1 on hypercholesterolemia-induced monopoiesis and its highly upregulated expression within aortic lesions, we propose TREM-1 as an attractive myeloid cell-specific target for attenuating the chronic inflammatory process in atherosclerosis.

## Discussion

The innate immune response-amplifying receptor TREM-1 was initially recognized for its central involvement in acute microbial driven diseases. However, emerging evidence also strongly implicates a role for TREM-1 in non-infectious inflammatory disorders, including, rheumatoid arthritis[32], psoriasis[33], acute pancreatitis[34], obstructive nephropathy[35] and pulmonary disease[36], hepatocellular carcinoma[29], lung[55] and colorectal cancer[37], as well as cardiac allograft rejection[38] and myocardial infarction[39].

Here, we provide evidence that TREM-1 is expressed in murine and human atherosclerotic lesions in situ and that TREM-1 potently exacerbated diet-induced atherogenesis. The pro-atherogenic impact of TREM-1 was not related to effects on systemic lipid or glucose metabolism. In contrast, our study identifies two previously unanticipated roles of TREM-1 in regulating myelopoiesis and cellular cholesterol metabolism. In particular, TREM-1 aggravated the dyslipidemia-induced peripheral blood monocytosis by promoting skewed monocyte differentiation from BM haematopoietic stem and progenitor cells. Numerous studies have established the link between gradually developing monocytosis and time-dependent lesion progression in atherosclerosis. The effects of TREM-1 on the peripheral monocyte pool only became significant upon prolonged HFCD feeding. In line with this observation, we did not

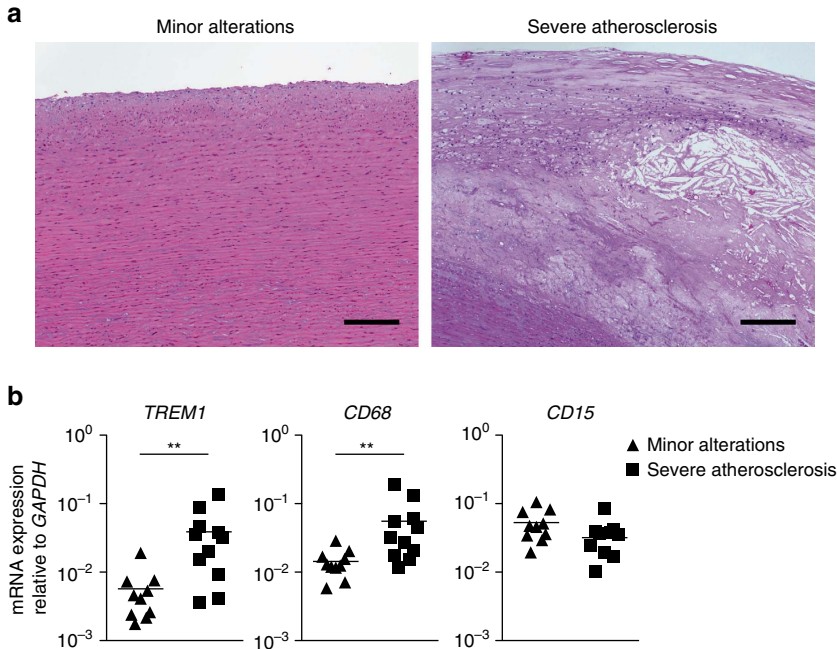

**Figure 7 | TREM1 is expressed in human atheromas.** (**a**) Representative hematoxylin and eosin stainings of sections from human aortas with either minor alterations (left) ($n = 10$) or severe atherosclerosis (right) ($n = 11$). Scale bars, 200 µm. (**b**) RNA was isolated from tissue rolls derived from FFPE human aortic specimens with either only minor alterations (diamonds) or severe atherosclerosis (squares). Expression of the indicated genes was assessed by qRT–PCR. Symbols represent values for individual patients, lines indicate mean expression. $**P < 0.01$ as determined by the two-tailed t-test.

**Figure 6 | TREM-1 promotes foam cell formation of human monocytes in vitro.** (**a,b**) TREM-1 and DAP12-expressing U937-TD cells were incubated for 48 h in vitro in the presence of 5% serum from HFCD-fed mice and the indicated stimuli (plate-bound anti-TREM-1, or isotype control antibody $+/- 30$ ng ml$^{-1}$ LPS). (**a**) Representative photomicrograph of ORO and DAPI-stained U937-TD cells. Scale bars indicate 100 µm. (**b**) Quantification of foam cell formation. The ratio of DAPI positive pixels versus ORO positive pixels was calculated using Image J software. Bars represent mean values + s.d. from 10 independent experiments. (**c,d**) U937-TD cells were incubated for 48 h with the indicated stimuli in the presence or absence of 5% HFCD serum. CD36 surface expression was determined by flow cytometry. (**c**) Representative histogram overlays showing CD36 surface expression (filled histograms represent isotype control-stained cells) (**d**) MFI values for CD36 surface expression. Bars show mean values + s.d. from three independent experiments. (**e–g**) CD14$^{hi}$ monocytes were flow-sorted from human blood donors. (**e,f**) Foam cell formation capacity of human CD14$^{hi}$ monocytes was determined as described for U937-TD cells. (**e**) Scale bars, 100 µm. (**f**) Bars show mean values + s.d. from three independent experiments with different blood donors. (**g**) After 20 h of culture with the indicated stimuli, CD14$^{hi}$ monocytes were harvested for qRT–PCR-based analysis of genes involved in cholesterol metabolism. Symbols show expression levels for $n = 7$ independent experiments with different blood donors. $*P < 0.05$, $**P < 0.01$ as determined by the one-way ANOVA test (**b-f**) and the paired t-test (**g**). Statistically not significant differences with $P > 0.05$ are not indicated.

note differences in the extent of early atherosclerotic lesions in $Trem1^{+/+}$ $Apoe^{-/-}$ versus $Trem1^{-/-}$ $Apoe^{-/-}$ mice at 4 weeks post HFCD or in mice that were on chow diet only (Supplementary Fig. 1a–d). Moreover, in contrast to the significant impact of TREM-1 on the extent of atherosclerotic surface area in the aorta at 16 weeks post HFCD, this effect was not revealed by the analysis of the cross-sectional lesion area at the aortic root. We propose that the most likely explanation for this seeming discrepancy is in fact the mechanism by which TREM-1 influences atherogenesis. In particular, by exacerbating HFCD-induced monocytosis TREM-1 will likely enhance the progression of already established lesions. Since lesions at the aortic root are usually the first to develop and represent the most advanced lesions throughout the aortic tree, at 16 weeks post HFCD these lesions might have been too advanced to still reflect the effects of TREM-1 on lesion development.

Under homeostatic conditions and on an ApoE-sufficient ($Apoe^{+/+}$) background, deficiency in TREM-1 has no appreciable impact on haematopoietic processes[29,31]. Here, we confirm and extend these data by showing that colony-forming capacity and colony subtype differentiation were comparable for $Trem1^{+/+}$ $Apoe^{-/-}$ versus $Trem1^{-/-}$ $Apoe^{-/-}$ mice under chow diet feeding. Strikingly, the aggravated monocytosis in HFCD-fed $Trem1^{+/+}$ $Apoe^{-/-}$ mice was not related to an increased expansion of haematopoietic stem or progenitor cells. Instead, we found that HFCD-feeding of $Trem1^{+/+}Apoe^{-/-}$ mice significantly skewed myeloid differentiation in $lin^-$ BM cells towards increased monocyte production. Since GMP express distinct levels of surface TREM-1 (Supplementary Fig. 2c)[31], this suggested a potential cell-autonomous fashion by which TREM-1 could regulate monocytic over granulocytic lineage specification. However, unlike what was observed for peripheral blood myeloid cells, HFCD feeding did not further increase surface TREM-1 expression on GMP (Supplementary Fig. 2c). Moreover, although ex vivo isolated GMP from $Trem1^{+/+}$ $Apoe^{-/-}$ mice exhibited increased mRNA levels for Irf8, a key transcription factor for monocytic lineage differentiation[42], stimulation of these GMP with plate-bound anti-TREM-1 in the presence or absence of HFCD serum was not sufficient to augment monocyte differentiation in vitro (Supplementary Fig. 2d,e). Whereas TREM-1 has been demonstrated a potent amplifier of cytokines such as M-CSF, GM-CSF, IL-6 and CCL2 (ref. 56), expression levels of these cytokines in the serum or bone flushes from HFCD-fed $Trem1^{+/+}$ $Apoe^{-/-}$ mice were generally close to the detection limit (Supplementary Fig. 6a,b). Hence, the question whether the HFCD-induced effect of TREM-1 on monopoiesis may be cell-intrinsic or extrinsic could not be conclusively addressed within the scope of the present study. Clearly, the complex contribution of TREM-1 to monopoiesis needs to be deciphered in further investigations that can take into account the multifarious dyslipidemia-induced changes in dietary PAMPs and cytokines in vivo.

While our findings show that TREM-1 can contribute to atherosclerosis through regulation of the BM and peripheral monocyte pool, resulting in increased monocyte/macrophage accumulation and lesion progression, our data also suggest that TREM-1 may promote atherosclerotic lesion progression in situ. In parallel with the increased abundance of monocyte/macrophage foam cell-associated transcripts, expression of Trem1 was highly upregulated in the aortic arch of 16 week HFCD-fed $Trem1^{+/+}$ $Apoe^{-/-}$ mice. TREM-1 surface expression on lesion-infiltrating cells could further be confirmed by flow cytometry. Moreover, major differences in the composition of $CD11b^+$ cells in digested aortas of $Trem1^{+/+}$ $Apoe^{-/-}$ versus $Trem1^{-/-}$ $Apoe^{-/-}$ mice could be attributed to $Ly6C^-$ macrophages but not to $Ly6C^{hi}$ monocytes or neutrophils,

arguing against a primary defect in monocyte recruitment in $Trem1^{-/-}$ $Apoe^{-/-}$ mice.

The presence of two distinct $Ly6C^-$ $CD64^+$ $MHCII^+$ versus $Ly6C^-$ $CD64^+$ $MHCII^-$ aortic macrophage populations is intriguing, in particular, as $MHCII^-$, but not $MHCII^+$, macrophages expressed TREM-1 and were more abundant in aortas of $Trem1^{+/+}$ $Apoe^{-/-}$ mice. To date, relatively few studies have performed detailed flow cytometry characterizations of aortic macrophage subsets. The existence of $CD11b^+$ $F4/80^+$ $MHCII^{lo}$ macrophages has been described for the steady-state aorta[44] and for the heart where multiple embryonic and adult macrophage subsets have been identified with different signatures for $MHCII^{hi}$ versus $MHCII^{lo}$ macrophages[57]. A recent study has suggested a similarly complex picture and multilayered origin for resident arterial macrophages[12]. As under homeostatic conditions, arterial macrophage populations appear to be sustained largely by self-renewal[12], yet recruitment of $Ly6C^{hi}$ monocytes is critically associated with foam cell formation in atherosclerosis[2,3,8,9], it is difficult to link the $MHCII^+$ and $MHCII^-$ macrophage populations identified in our study with the macrophage subsets described by Ensan et al.[12]. Our attempts to further characterize aortic macrophages in $Trem1^{+/+}$ $Apoe^{-/-}$ versus $Trem1^{-/-}$ $Apoe^{-/-}$ mice were hampered by the very low cell yield, which precluded robust gene expression analysis in sorted cell subsets. Hence, we can currently only speculate about the origin and functional profile of the aortic $MHCII^-$ $TREM-1^+$ versus $MHCII^+$ $TREM-1^-$ macrophages. Based on findings from experimental colitis models, where the expression pattern of TREM-1 has already been characterized, TREM-1 is expressed by infiltrating $Ly6C^{hi}$ effector monocytes and their $Ly6C^{lo}$ descendants but not by resident $Ly6C^-$ $MHCII^+$ macrophages in the inflamed colon[47]. Moreover, during intestinal inflammation, regular differentiation of $Ly6C^{hi}$ monocytes to $Ly6C^-$ $MHCII^+$ macrophages is arrested at the intermediary stage of $Ly6C^{int}$ $MHCII^+$ macrophages with pro-inflammatory properties[45–47]. In aortas of HFCD-fed $Trem1^{+/+}$ $Apoe^{-/-}$ mice, highest TREM-1 MFI values among infiltrating $Ly6G^-$ $CD11b^+$ cells were observed in $Ly6C^{int}$ $MHCII^+$ macrophages. While it remains to be determined whether $Ly6C^{int}$ $MHCII^+$ cells give rise to $Ly6C^-$ $MHCII^-$ $TREM-1^+$ macrophages in vivo, our preliminary data indicate a distinct down-modulation of HLA-DR after differentiation of human monocytes in the presence of M-CSF and agonistic anti-TREM-1 stimulation in vitro (Supplementary Fig. 7a,b). At least an initial $Ly6C^{hi}$ monocytic origin of $Ly6C^{int}$ $MHCII^+$ $TREM-1^+$ macrophages appears conceivable also for the aorta[2,3,8]. In mice with experimental colitis, TREM-1 is barely expressed on peripheral blood $Ly6C^{hi}$ monocytes[47], and although in HFCD-fed $Trem1^{+/+}$ $Apoe^{-/-}$ mice TREM-1 expression was upregulated on all peripheral blood cell subsets, TREM-1 surface expression on $Ly6C^{hi}$ monocytes remained low. Hence, TREM-1 must become upregulated on infiltrating monocytes following transmigration to inflamed tissues. Whereas in the colitic intestinal mucosa, TREM-1 is likely induced by microbial factors[27,58,59], in the subendothelial space of aortic vessels a role for non-microbial DAMPs appears more probable as we observed potent upregulation of TREM-1 under dyslipidemic conditions in vivo and upon stimulation of primary monocytes with HFCD serum or purified oxLDL in vitro. Besides oxLDL or dietary DAMPs also GM-CSF secreted by endothelial cells could mediate local induction of TREM-1 (refs 15,43).

The factors that may engage and activate TREM-1 in aortic lesions are more difficult to grasp, in particular, as TREM-1 ligands remain incompletely defined. In contrast to other TREM family members, TREM-1 does not appear to participate in recognition of lipids[60,61]. Nonetheless, the recently described

capacity of the PAMP molecule HMGB1 (refs 28,29) and multimerized PGLYRP1 (ref. 30) to associate with and stimulate TREM-1 provides a possible scenario for activation of lesion-associated myeloid cells in the context of non-infectious inflammation. PGLYRP1 is expressed by neutrophils which seem abundant enough in murine lesions. Moreover, HMGB1 is released from macrophage foam cells that accumulate in advanced atheromas[18,19]. An HMGB1-mediated local triggering of TREM-1 in concert with the progressive monocytosis in $Trem1^{+/+} Apoe^{-/-}$ mice could provide an explanation for the impact of TREM-1 on lateral lesion growth rather than early lesion initiation. It is also tempting to speculate that TREM-1, possibly in association with other receptors such as RAGE, may be involved in the recently reported priming of lesional macrophages by neutrophil extracellular traps[62].

A striking finding of the present study was the potent effect of TREM-1 on foam cell formation *in vitro*. Although innate immune signalling pathways have previously been acknowledged to antagonize compensatory cholesterol efflux mechanisms[63], the direct impact of TREM-1 on foam cell conversion was unanticipated, considering that TREM-1 on its own otherwise has poor stimulatory capacity[25]. TREM-1-mediated stimulation of U937-TD cells but also of primary human monocytes associated with a pronounced upregulation of the oxLDL scavenger receptor CD36. The TREM-1-induced expression of CD36 is consistent with the augmented expression of PPARγ as a TREM-1-regulated gene[56] and the role of PPARγ as key regulator of CD36 expression and foam cell formation[64,65]. It is further noteworthy that in the converse expression of molecules on mouse and human monocyte subsets TREM-1 segregates with PPARγ and CD36 and that both TREM-1 and CD36 are upregulated during steady-state differentiation of Ly6C$^{hi}$ to Ly6C$^{lo}$ monocytes[66,67]. Moreover, CD36 has been ascribed a central role in facilitating TLR4/6 assembly leading to increased pro-inflammatory cytokine and ROS production[24]. Notably, in primary human monocytes TREM-1 not only augmented mRNA expression of *PPARG* and *CD36* but also of other receptors implicated in lipid uptake such as *MSR1* and *LDLR*. Whether the clear impact of TREM-1 on several other genes involved in intracellular cholesterol metabolism represents a direct effect of

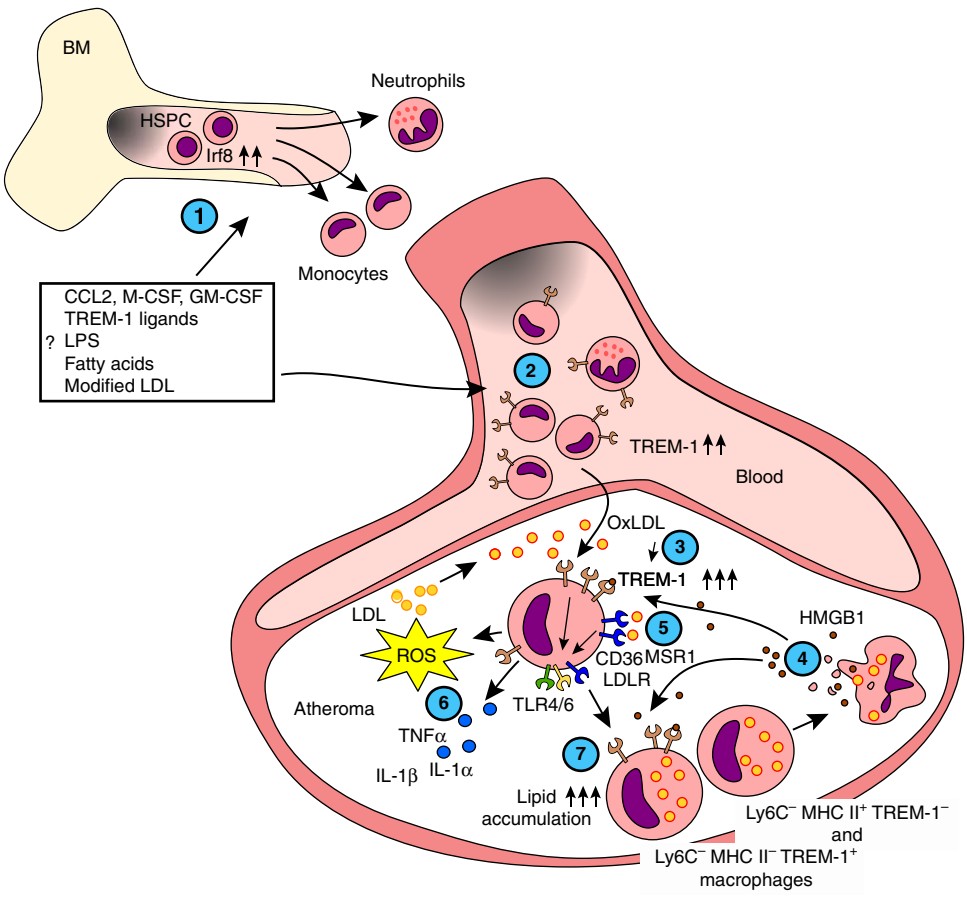

**Figure 8 | Model for multilayered pathogenic roles of TREM-1 in diet-induced atherogenesis.** (**1**) TREM-1 skews myeloid differentiation towards increased monocyte output by yet undetermined direct or indirect mechanisms, likely acting together with microbial or dietary PAMPs which become systemically available following HFCD feeding. (**2**) These factors may also be involved on the upregulation of TREM-1 on peripheral blood myeloid cell subsets. (**3**) TREM-1 is further upregulated on Ly6C$^{hi}$ monocytes following their transmigration to the intima and exposure to oxLDL. (**4**) Activation of monocytes via TREM-1 may be mediated by neutrophil-derived PGLYRP1 (not depicted) or by HMGB1 released by activated or necrotic macrophages. (**5**) TREM-1-mediated activation promotes the expression of scavenger receptors CD36, MSR1 and the LDLR and thereby increases lipid uptake. TREM-1 further promotes foam cell formation by interfering with intracellular cholesterol transport and efflux mechanisms (not depicted). (**6**) CD36 facilitates TLR4/6 assembly and NLRP3 activation (not depicted) leading to increased pro-inflammatory cytokine and ROS production, which may contribute to additional lipid peroxidation. Concurring TREM-1-mediated activation could further amplify these pro-inflammatory processes. (**7**) Increased uptake of oxLDL leads to the differentiation of Ly6C$^{hi}$ monocytes to Ly6C$^-$ macrophage foam cells through a Ly6C$^{int}$ TREM-1$^+$ MHCII$^+$ intermediary stage.

TREM-1 or compensatory response to the increased lipid content remains to be determined.

In support of our hypothesis that TREM-1 may promote atherosclerotic lesion progression locally, we found highly upregulated expression of mRNA for TREM-1 also in human aortic tissue specimens displaying severe atherosclerosis. Thus, it is tempting to speculate that the proposed association of distinct *TREM1* polymorphisms with coronary artery disease in a Russian population may be causal[54]. In the Swiss population-based CoLaus study[68], an association of the *TREM1* single-nucleotide polymorphism (SNP) rs2234237 with self-reported cardiovascular disease could not be established. Nonetheless, in a substudy of CoLaus, participants with the TT genotype had significantly lower intima-media thickness (IMT) maximal values for the left carotid artery compared with AA or AT genotype carriers, with a significant additive effect ($P < 0.006$ for trend); a similar, albeit non-significant, trend was found for the right carotid artery (Supplementary Table 3). Therefore, the potential significance of TREM-1 for human cardiovascular disease clearly merits more attention.

In summary, while the precise mechanisms behind TREM-1-mediated lesion progression remain to be further determined, we propose a multiphasic model for TREM-1 in atherogenesis (Fig. 8): In the periphery, TREM-1-mediated signalling skews myelopoiesis towards increased production of monocytes. In atherosclerotic lesions, TREM-1 participates in a positive feedback-loop with oxLDL and CD36, thus contributing to increased lipid uptake and foam cell formation. This process may additionally be boosted by multiple adverse effects of TREM-1 on cellular lipid uptake, transport and efflux mechanisms as well as by TREM-1-amplified lipid peroxidation and pro-inflammatory cytokine production in concert with TLR4/6. Our findings thus identify TREM-1 as a key innate immune receptor participating in the chronic inflammatory process in atherosclerosis.

## Methods

**Mice.** $Trem1^{-/-}$ mice (C57BL/6) were generated and characterized in our laboratory[31]. $Apoe^{-/-}$ (B6.129P2-$Apoe^{tm1Unc}$/J) mice (C57BL/6), were purchased from Charles River Laboratories (USA). $Trem1^{-/-}$ mice were backcrossed with $Apoe^{-/-}$ mice to generate $Trem1^{+/+}$ $Apoe^{-/-}$ and $Trem1^{-/-}$ $Apoe^{-/-}$ mice. Mice were maintained in isolated ventilated cages under specific pathogen-free conditions. Fourteen days before and every 14 days throughout feeding experiments, bedding between $Trem1^{+/+}$ $Apoe^{-/-}$ and $Trem1^{-/-}$ $Apoe^{-/-}$ mice was exchanged to adjust for potential differences in microbiota composition. All experimental procedures were approved by the Veterinary Office of the Canton of Bern (license no: 111/14) and performed in compliance with Swiss laws for animal protection.

**Induction of experimental atherosclerosis.** Six- to eight-week-old female $Trem1^{+/+}$ $Apoe^{-/-}$ and $Trem1^{-/-}$ $Apoe^{-/-}$ mice were either further maintained on a regular chow diet or placed on a high-fat, high cholesterol diet (HFCD) (D12007C, Research Diet Inc., New Brunswick, NJ USA) for 4-16 weeks as indicated.

**Determination of atherosclerotic lesion size.** Following euthanasia by $CO_2$-exposure, mice were perfused for 5 min with ice-cold PBS and subsequently with 4% buffered paraformaldehyde (PFA) containing 2 mM EDTA and 2% glucose. Whole aortas including the initial parts of the brachiocephalic artery, the left common carotid artery, the left subclavian artery the right and left iliac arteries were carefully freed from all connective tissue and fat, were cut open longitudinally and pinned 'en face' on petri dishes coated with activated carbon blackened paraffin. Atherosclerotic lesions were visualized by staining in 0.3% Oil-Red-O (Sigma, Switzerland) staining solution in 60% isopropanol. The aortic surfaces were photographed (D90, Nikon) and the lesion sizes were measured using ImageJ software (version 2.0.0-rc-43/1.51c). The extent of atherosclerosis was determined by calculating the ratio of the lesion-covered surface area compared to the total surface area of the aorta.

To analyse cross-sections of the aortic root, hearts were cut in half and the upper part was embedded in O.C.T. (TissueTek, Sakura Finetek Europe, Leiden, NL). Cross-sections (6 μm thick) were prepared from the frozen hearts on a Leica CM1950 cryostat (Leica Biosystems Germany). Ten cross-sections of the aortic root were collected every 36 μm, stained with ORO and counterstained with Mayer's hematoxylin staining solution. Sections were analysed on a Zeiss Axiophot 2 (Zeiss, Germany) and the lesion area was calculated using ImageJ software in a blinded manner.

**Analysis of serum lipid and soluble TREM-1 (sTREM-1).** Serum was obtained from whole blood collected from the '*vena cava*' of o/n fasted mice. Total cholesterol, HDL and LDL cholesterol were determined by the Department of Clinical Chemistry of the University of Bern on a cobas 8000 clinical chemistry analyser (Roche diagnostics, Switzerland). sTREM-1 was determined by ELISA (R&D Systems, USA).

**Nanostring gene expression profiling of aortic RNA.** Aortas were prepared as described above but without PFA perfusion. Aortas were cut in four pieces consisting of the aortic arch, descending thoracic aorta, abdominal aorta and the iliac bifurcation and stored in RNAlater (Life Technologies, USA). For RNA isolation, aortic pieces were transferred into TRI Reagent (Sigma-Aldrich, Switzerland) and tissues were homogenized with a Qiashredder (Qiagen) $2 \times$ at 20 Hz for 5 min. RNA was then isolated according to the manufacturer's protocol. For mRNA expression profiling, the nCounter Mouse Immunology panel (Nanostring technologies, Seattle, USA) was complemented with additional 30 genes of interest (Supplementary Table 1). Expression levels of mRNA for TREM-1 and other TREM family members were verified by SYBR-green (Roche, Switzerland) based qRT-PCR analysis using QuantiTect primer assays (Qiagen) and normalization to *Gapdh* (Mm_Gapdh, QT01658692; Mm_Trem1, QT00153979; Mm_Cd68, QT00254051; Mm_Ly6g, QT00529655).

**Isolation of cells from murine tissues.** Peripheral blood was collected from the '*vena cava*' post mortem or from the lateral tail vein from live animals during HFCD feeding. Single-cell suspensions from spleen were prepared by the enzymatic digestion protocol used for aortic cell preparation as described below. For BM cells, tibiae and femurs were placed in RPMI 1640 medium supplemented with 10% FCS and flushed with a 26 GA 3/8 needle. Peripheral blood, spleen and BM cells were erythrocyte-depleted with Tris-buffered ammonium chloride. For preparation of aortic single-cell suspensions, aortas were carefully dissected and freed from connective tissue. Aortas were incubated in HEPES-buffered saline containing 5% horse serum as well as collagenase I (1,350 U ml$^{-1}$, Sigma), Collagenase XI (375 U ml$^{-1}$, Sigma), Hyaluronidase type I-s (180 U ml$^{-1}$, Sigma) and DNase I (180 U ml$^{-1}$, Roche) for 100 min at 37 °C with magnetic stirring. Digested aortas were filtered through 45 μM cell strainers (Falcon Technologies, BD) to obtain single-cell suspensions.

**Flow cytometry characterization of murine cell subsets.** Peripheral blood and aortic single-cell suspensions were Fc-blocked and stained with the following monoclonal antibodies (mAbs) (Biolegend, San Diego, USA, unless indicated otherwise): anti-CD45-Pacific Blue (30-F11; 0.25 μg ml$^{-1}$), anti-CD11b-PE-Cy7 (M1/70; 0.5 μg ml$^{-1}$), anti-MHC class II-APC-Cy7 (M5/114.15.2; 0.067 μg ml$^{-1}$), anti-CD11c-Alexa Fluor 700 (N418; 2.5 μg ml$^{-1}$), anti-Ly6C-FITC (AL-21; 0.5 μg ml$^{-1}$), anti-F4/80-PE-Cy7 (BM8, 0.5 μg ml$^{-1}$) (BD Pharmingen, USA), anti-Ly6G-PerCP-Cy5.5 (1A8; 0.2 μg ml$^{-1}$), anti-CD64-PE (X54-5/7.1; 2 μg ml$^{-1}$), anti-TREM-1-eFluor 660 (TR3MBL1; 0.2 μg ml$^{-1}$) (eBioscience, USA). Following matched fluorochrome labelled isotype control antibodies were included: for CD11c: armenian hamster Ig (HTK888); for CD64: mouse IgG1(MOPC-21) (BD Pharmingen); for TREM-1: rat IgG2a (eBR2a) (eBioscience). BM cells were lineage-depleted with MACS (Miltenyi Biotech) using anti-CD19-biotin (6D5; 1.67 μg ml$^{-1}$), anti-CD3e-biotin (145-2C11; 1.67 μg ml$^{-1}$), anti-Gr1-biotin (RB6-8C5; 1.67 μg ml$^{-1}$) anti-Ter119 (Ter119; 1.67 μg ml$^{-1}$) (Biolegend) and anti-biotin Microbeads (Miltenyi Biotec). Lineage-depleted cells were stained for HSC and myeloid progenitors with anti-CD127-PE (SB/199; 2 μg ml$^{-1}$), anti-CD117-APC-Cy7 (2B8; 0.67 μg ml$^{-1}$), anti-CD16/32-PE-Cy7 (93; 0.5 μg ml$^{-1}$) (Biolegend) and anti-Ly-6A/E (Sca-1)-PerCP-Cy5.5 (D7; 0.33 μg ml$^{-1}$), anti-CD34-eFluor450 (RAM34; 2 μg ml$^{-1}$) (eBioscience). For all cell acquisitions, live-dead exclusion was performed using DAPI (Sigma; 0.5 μg ml$^{-1}$) or LIVE/DEAD fixable blue dead cell stain kit (Thermo Fisher Scientific, USA). Cells were acquired on a LSRII (BD Biosciences, USA) and flow cytometry data were analysed using FlowJo version 9.9 × or version 10.1 ×.

**Myeloid colony-forming unit assays.** A total of $5 \times 10^3$ lin$^-$ BM cells were cultured in MethoCult base medium (STEMCELL Technologies, Grenoble, France) supplemented with 15% FCS, 20% BIT (50 mg ml$^{-1}$ BSA in Iscove's modified Dulbecco's medium, 1.44 U ml$^{-1}$ rh-insulin (Actrapid, Novo Nordisk, Denmark), and 250 ng ml$^{-1}$ human Holo Transferrin [ProSpec, USA]), 100 μM 2-β-mercaptoethanol, 100 U ml$^{-1}$ penicillin, 100 μg ml$^{-1}$ streptomycin, 2 mM L-glutamine, and a cytokine mix of 50 ng ml$^{-1}$ rm-SCF, 10 ng ml$^{-1}$ rm-IL-3, 10 ng ml$^{-1}$ rh-IL-6, and 50 ng ml$^{-1}$ rm-Flt3-ligand (all from ProSpec, USA). Colonies were enumerated after 7 days of culture. Colony subtypes were

determined by inverted light microscopy and by Pappenheim staining of individually plucked colonies.

**Analysis of human peripheral blood CD14^hi monocytes.** Buffy coats were obtained from the local blood donation center. PBMCs were purified by Ficoll-Paque PLUS gradient centrifugation (GE Healthcare, UK) and stained for FACS purification with anti-human CD14-Alexa Fluor 488 (HCD14; 0.4 µg ml$^{-1}$) and CD16-Pacific Blue (3G8; 0.1 µg ml$^{-1}$) (Biolegend). Sorted CD14$^{hi}$ monocytes were cultured at 100,000 cells per well in 96-well U-bottom plates pre-coated with 10 µg ml$^{-1}$ agonistic anti-TREM-1 antibody (MAB1278) (R&D Systems) or an isoype control antibody (MOPC21) (Biolegend). Where indicated, the culture medium (RPMI 1640 with 10% FCS) was supplemented with either 30 ng ml$^{-1}$ LPS (Sigma), 50 µg ml$^{-1}$ endotoxin-free oxLDL (Biomedical Technologies, Italy) or with 5% heat-inactivated murine serum from 16-week HFCD-fed $Trem1^{+/+}$ $Apoe^{-/-}$ mice. Following 20 h in vitro culture, supernatants were harvested and used for TNFα, IL-1α and IL-1β ELISAs (Biolegend) according to the manufacturer's protocols. In some experiments, monocytes were harvested into TRI Reagent for RNA isolation and qRT-PCR of cholesterol-metabolism associated genes using QuantiTect primer assays (Qiagen). (Hs_CD36: QT00020181, Hs_MSR1: QT00064141, Hs_LDLR: QT00045864, Hs_ABCA1: QT00064869, Hs_ABCG1: QT00021035, Hs_PPRG: QT00029841, Hs_NPC1: QT00066465, Hs_NPC2: QT00064904, Hs_STARD4: QT00015925, Hs_LIPA: QT00021679, Hs_GAPDH: QT00079247). For flow cytometry, monocytes were detached by incubation with Accutase (Innovative Cell Technologies, USA) and stained with anti-human TREM-1-APC (TREM-26; 0.25 µg ml$^{-1}$) or a matched isotype control antibody (MOPC-173) (Biolegend). Foam cell formation capacity of human monocytes was analysed after 48 h of culture as described below for U937-TD cells.

**Culture of U937-TD cells and foam cell formation in vitro.** U937 myelomonocytic cells stably transduced with human TREM-1 and DAP12 (U937-TD) were generated by Tessarz et al.[53]. U937-TD cells were cultured in RPMI 1640 supplemented with 10% FCS. For foam cell formation assays and analysis of CD36 surface expression, U937-TD cells were cultured at 75,000 cells per well in 96-well U-bottom plates pre-coated with 10 µg ml$^{-1}$ agonistic anti-TREM-1 antibody (MAB1278) or an isotype control antibody (MOPC21) in the presence or absence of 30 ng ml$^{-1}$ LPS (Sigma) or 5% heat-inactivated serum obtained from 16-week HFCD-fed $Trem1^{+/+}$ $Apoe^{-/-}$ mice. After 48 h of culture, cells were gently detached by Accutase (Innovative Cell Technologies) and stained with anti-human CD36-APC (5–271; 0.125 µg ml$^{-1}$) (Biolegend) for flow cytometry. Alternatively, 100,000 cells per 100 µl were cytospun for 1 min at 800 r.p.m. onto Superfrost Plus microscope slides (Thermo Scientific). After fixation in 4% PFA in PBS, slides were stained with 0.3% Oil-Red-O staining solution in 60% isopropanol for 30 min, washed with PBS and counterstained with DAPI (Sigma; 0.5 µg ml$^{-1}$). Images were acquired on a Olympus IX81 confocal microscope at 200× magnification. The ratio ORO:DAPI was assessed by calculating the respective total pixels with ImageJ software.

**Human aortic tissue specimens.** Human FFPE aortic tissue specimens were collected at the Institute of Pathology, University of Bern between 2011–2013 with the written consent of patients and the approval by the Ethics Committee Bern, Switzerland. Suitable specimens were selected by an expert pathologist (Y.B.) and based on histological criteria from the analysis of H&E stained tissue sections categorized into either specimens with minor alterations or with severe atherosclerotic plaques. RNA from five 10 µm thick FFPE tissue roles was purified using a column-based approach described by Oberli et al.[69]. SYBR-green (Roche) based qRT-PCR was performed using QuantiTect primer assays (Qiagen) (Hs_TREM1: QT00046284, Hs_CD68: QT00037184, Hs_FUT4(CD15): QT00223657).

**CoLaus substudy and intima-media thickness (IMT) measurements.** Clinical characteristics of the CoLaus[68] substudy participants according to the TREM1 SNP rs2234237 genotype are provided in Supplementary Table 4. IMT was assessed using B-mode high resolution imaging of the carotid arteries. IMT values are expressed in mm.

**Statistical analyses.** All data except for Nanostring data and IMT data were analysed using Prism software version 5.0f. Statistical tests were used as indicated in the figure legends. In general, the two-tailed Mann–Whitney t-test was performed to compare two groups. One-way analysis of variance (ANOVA) test with Dunn's multiple comparison test was used to compare three or more groups with each other. Two-way ANOVA test with Bonferroni correction was used to compare multiple data groups with two independent variables. For analysis of the Nanostring data, significant differences in gene expression were calculated between the wild-type and knock-out groups of each age-diet group. Gene expression values were first log-transformed. The Bioconductor limma package was used to perform a moderated t-test on all genes[70]. Principal component analysis was done in R and the results were visualized with the ggplot2 package[71].

For IMT, between-group comparisons were performed using one-way ANOVA and the results were expressed as mean ± s.d. Multivariable analysis was performed using ANOVA adjusting for gender, smoking status (never, former, current), age (continuous) and body mass index (continuous) and the results were expressed as adjusted mean ± s.e. Genetic effect was assessed supposing an additive effect and applying the qtlsnp procedure of Stata. Statistical analysis was conducted using Stata version 13.1 (Stata Corp).

**Study approval.** Patients and blood donors gave written informed consent. Analyses of human formalin-fixed paraffin-embedded tissue specimens was approved by the Ethics Committee Bern, Switzerland.

**Data availability.** All relevant data that support the findings of this study, including the full Nanostring gene expression profiling data, are available from the corresponding authors on request.

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

## Acknowledgements

We thank Murielle Bochud and Gérard Waeber from the University of Lausanne for their analyses of the *TREM1* SNP data in the CoLaus cohort and their time for discussions. We further thank the FACS sorting facility of the Department of Clinical Research, University of Bern for excellent assistance with cell sorting. We are very grateful to Anne-Laure Huguenin and Ursina Lüthi from the Department of Clinical Research, University of Bern for their expert help with the progenitor isolation and CFU assays. This work was supported by grants from the Swiss National Science Foundation (no. 138392 to C.M.) and from the Novartis Foundation for medical-biological research (no. 13C173 to L.S.).

## Author contributions

D.Z. planned and conducted the majority of experiments with the assistance of S.R. and J.B. B.W. was involved in generation of the *Trem1*⁻/⁻ mice and initial experiments. S.F. and C.R. critically contributed to the design of the study with support from A.F.O. Y.B. histopathologically assessed human aortic tissue specimens. A.C. generated the U937-TD cells. C.S. and P.M.-V. performed statistical analyses of the Nanostring and *TREM-1*-SNP/IMT data, respectively. L.S. and C.M. designed the study and wrote the manuscript together.

## Additional information

**Competing financial interests:** The authors declare no competing financial interests.

**How to cite this article**: Zysset, D. *et al.* TREM-1 links dyslipidemia to inflammation and lipid deposition in atherosclerosis. *Nat. Commun.* **7**, 13151 doi: 10.1038/ncomms13151 (2016).

