## [Peer Review File · Nature Communications]

Reviewers' comments:

Reviewer #1 (expert in atherosclerosis)

Remarks to the Author:

This is an interesting paper that illustrates the role of TREM-1 in regulating macrophage function in atherosclerosis. The manuscript is well-written and the experiments have appropriate controls. Some details are missing that are listed below.

1--Figure 1: the aorta images in panel A do not seem to accurately reflect the 50% reduction in atherosclerosis; the images seem to be ~70%. Please use a better representative image for the cumulative data.

2--Figure 1. The glucose data do not add to the paper.

3--Figure 2--complete flow cytometry gating strategies should be shown in the supplement, for monocytes and their upstream progenitors. It is unclear what cells are being gated on here in these bar graphs.

4--Figure 2: representative images of the cells obtained from the methylcellulose culture should be shown.

5--Please explain rationale for why RNA-Seq of whole aorta was performed rather than performing RNA-Seq on macrophages from aorta. These data are confusing as multiple cell types can express many of these genes.

6--the data in Figure 4 are uninterpretable based on the FACS gating shown. Please show full gating strategy and explain as this is an unconventional way to identify these cells.

Reviewer #2 (expert in immunology)

Remarks to the Author:

Here the authors have identified a previously unknown effect of TREM1 in atherosclerosis. They demonstrate that TREM1 is expressed in advanced human atheroma and up-regulated on peripheral and lesional myeloid cells in the apoe^{-/-} mice model. Furthermore, they propose that TREM1 up-regulation correlates with hyperlipidemia induced monocytois and that it synergizes with serum factors (currently undefined) to exacerbate foam cell formation. Lastly, they report a skewed monocyte differentiation pattern and proposed CD36 up-regulation as a mechanism by which TREM1 exacerbate lesion progression. Overall the authors suggest a two phase model where TREM1 promotes monocyte production during HFCD feeding and then promotes Foam cell production, resulting in larger plaques.

This is an intriguing model. Unfortunately much of the data recapitulate known functions of TREM1 such as promotion of inflammatory cytokine production, and differenced in intensity on PMN and macrophages, and up-regulation in inflammatory settings. Here we are, therefore, left with interesting new effects on hematopoiesis that are completely unexplained, and new effects on foam cell production that are also unexplained. The macrophage population stuff is largely descriptive and does not help to understand how TREM1 affects disease. Additional information here would strengthen the

paper substantially

Additional Major concerns:

1- Since the histological evaluation of atherosclerotic lesions in the apoe^{-/-} mice model was the foundation for the subsequent findings, evaluation of the aortic sinus at 16 weeks post HFCD is critical to fully understand the mechanism by which TREM1 regulates lesion formation. This anatomical site was evaluated at an earlier time point (4 weeks post HFCD) but not in the advance stage. Moreover if as suggested monocyctosis and foam cell formation are promoted by TREM1, why is this effect not seen at 4 weeks?

2- To properly characterize the in vivo effects of TREM1 on vascular inflammatory cells the gene expression profiling in the aorta should have been normalized to a monocyte/macrophage marker to account for macrophage lesion and infiltration differences already described. Although the authors argued they restricted the analysis to the aortic arch, where " the least differences in total lesion size was observed", a bias is clear due to up-regulation of common lesional macrophage markers (figure 3D). Furthermore, why does TREM1 does not come up in the gene profiling analysis further validating the previous qPCR data (figure 3A).

Minor concerns:

1- The statistical analysis seems unclear. In many cases, points with wildly overlapping error bars are denoted as significantly different. It does not seem that that could be the case. See 2a and 5a

2- Since the authors suggest that TREM1 signaling induces monocytic skewing of hematopoiesis in the bone marrow, this would suggest that these progenitors express TREM1? Is this the case?

3- TREM1^{-/-} should be shown in 2F

4- The gene expression data does not seem to offer anything to the paper. If fact, CD36 is shown to be not different yet this is pursued in the paper

5- Soluble TREM1 is not shown at all. Does HFCD yield sTREM1? Do patients with advanced disease have high sTREM1 levels?

6- Other members of the TREM family have been or are likely involved in atherosclerosis. Specifically, Trem14 is known to be involved, Trem2 has been implicated and Trem1 is a platelet gene. Does the elimination of TREM1 affect the expression of any of these other genes?

7- Foam cell formation and the regulation of this process should be assessed in primary monocytes from humans.

Reviewer #3 (expert in atherosclerosis)

Remarks to the Author:

In the manuscript "TREM-1 links dyslipidemia to inflammation and lipid deposition in atherosclerosis" Zysset et al. profile development of atherosclerosis in Apoe^{-/-} Trem-1^{-/-} mice. They show that the DKO have smaller lesions without effects on lipids, and less monocyctosis. Many inflammatory genes are attenuated in the aortas of the DKOs, as demonstrated by gene expression profiling. TREM-1 is augmented on circulating monocytes after high fat diet and is expressed somewhat on monocytes and

neutrophils in the aorta. Mechanistically, the authors suggest that oxLDL upregulates TREM-1 and TREM-1 increases cytokine production. They also suggest that TREM-1 augments CD36 and foam cell formation. Finally, the authors show that TREM-1 is expressed in human lesions. Overall, this is a good paper.

1. The authors should show their gating strategy for the various HSPC as shown in Figure 2.
2. TREM-1 is only mildly augmented in the various cell subsets, as shown by flow cytometry in Figure 4. It would be reassuring to see TREM-1 expression on sorted cells by either PCR (for message) or on Western blots (for protein).

Point-by-point reply to Reviewers' comments on manuscript NCOMMS-16-05799 "TREM-1 links dyslipidemia to inflammation and lipid deposition in atherosclerosis"

The authors would like to thank all the Reviewers for their time, their careful reading of the manuscript and their insightful and highly constructive criticisms. The original Reviewers' comments have been copied in blue font color. Our answers are marked in black font color and changes in the revised manuscript have been indicated in red font color.

Reviewer #1 (expert in atherosclerosis)
Remarks to the Author:

This is an interesting paper that illustrates the role of TREM-1 in regulating macrophage function in atherosclerosis. The manuscript is well-written and the experiments have appropriate controls. Some details are missing that are listed below.

1--Figure 1: the aorta images in panel A do not seem to accurately reflect the 50% reduction in atherosclerosis; the images seem to be ~70%. Please use a better representative image for the cumulative data.

The previous Figure 1a showed examples of aortas with the *highest* and *lowest* extent of atherosclerosis for both groups of mice as indicated in the previous Figure legend. However, we understand that this approach may be confusing for the reader. In the revised version of the manuscript we have therefore included only one representative aorta picture for each group of mice to illustrate the *average* extent of atherosclerosis (revised Fig. 1a).

2--Figure 1. The glucose data do not add to the paper.

We have removed the glucose data from the revised Figure 1 and adjusted the text throughout the manuscript. Since the information on the equal glucose serum levels / clearance in *Trem1^{+/+} Apoe^{-/-}* vs. *Trem1^{-/-} Apoe^{-/-}* mice could still be of interest to a number of readers (e.g. considering the phenotype of *Tlr4^{-/-}* mice that are partially protected against high fat diet-induced insulin resistance e.g. Shi H et al., J Clin Invest, Nov 2006), we have moved the glucose data to a new Supplementary Figure 1 (Supplementary Fig. 1c-e) in the revised manuscript.

3--Figure 2--complete flow cytometry gating strategies should be shown in the supplement, for monocytes and their upstream progenitors. It is unclear what cells are being gated on here in these bar graphs.

We apologize for this unclarity. Representative gating schemes are now included in a new Supplementary Figure 2 (Supplementary Fig. 2a,b).

4--Figure 2: representative images of the cells obtained from the methylcellulose culture should be shown.

Representative images of monocytic and granulocytic colonies are now included in the revised Figure 2 (revised Fig. 2e).

5--Please explain rationale for why RNA-Seq of whole aorta was performed rather than performing RNA-Seq on macrophages from aorta. These data are confusing as multiple cell types can express many of these genes.

There are three main reasons to this approach: First, we aimed for an unbiased approach in order to screen for potential TREM-1-mediated (indirect) effects also outside the myeloid compartment. Second, the purification of a sufficient number of macrophage for robust RNA isolation and gene expression analysis would have required pooling cells from several aortas (our own experience and see also Ensan S et al., *Nat Immunol* 17,159-168 (2015) where "18-20 aortas were pooled per sample") and an enormous number of mice to generate multiple samples from independent experiments. This was not only beyond the scope of our facilities but could also have obscured findings based on intra-group variations in the extent of atherosclerosis. Last, with the enzymatic digestion protocol for release of aortic macrophages, a loss in RNA integrity and potentially biased data also had to be considered. In order to explain the difficulty of assessing gene expression in isolated macrophages to the reader, we have inserted an additional sentence into the corresponding paragraph of the Results section: p 7, lines 214 - 216.

6--the data in Figure 4 are uninterpretable based on the FACS gating shown. Please show full gating strategy and explain as this is an unconventional way to identify these cells.

We regret that our gating strategy for aortic cells appears confusing. However, except for the pre-gating of single, live, non-autofluorescent cells, the complete gating strategy has been shown in the previous Figure 4 and was also fully described in the Results section. As illustrated by two references included in the Results section, a very similar gating approach was also employed by others in the field (Choi JH et al., *Immunity* 35, 819-831 (2011); Ensan S et al., *Nat Immunol* 17,159-168 (2015), with the latter study showing that MHC class II⁺ CD11c⁺ cells indeed are dendritic cells based on their F4/80^{lo} and Zbt46^{hi} phenotype. Although F4/80 is commonly used for identification of macrophages, the study by Ensan et al. in fact also shows that (at least in the steady state) an F4/80-based pre-gating *a priori* excludes Ly6C^{hi} monocytes and that arterial macrophages in older mice are predominantly F4/80^{lo}. Since the F4/80 staining in our hands generally is rather unreliable / dim and we also wanted to visualize Ly6C^{hi} monocytes, we chose segregation of monocyte/macrophage subsets (and neutrophils) by plotting Ly6C against MHC class II (adapted from the "monocyte waterfall" model by Tamoutounour S et al.,

Eur J Immunol. 2012 Dec;42(12). The monocyte/macrophage origin was subsequently confirmed by analysis of the core macrophage marker CD64, which in our opinion is superior (i.e. unequivocal / brighter staining) to F4/80. As shown by Ensan et al., except for Ly6C^{hi/int} monocytes, all F4/80⁺ macrophages are indeed CD64⁺.

For the reasons stated above, we still consider our gating strategy a valid approach. Nonetheless, we agree that we should provide more transparency, and in a new Supplementary Figure 4 we now also show the initial gating for single, live, non-autofluorescent cells to make the sequence of gating more accessible to the reader. The data in Figure 4 are further taken up in the introductory paragraph of Figure 5 (p 9, lines 277 – 282) and discussed in detail in the Discussion section: p 14-15, lines 460 - 472.

Reviewer #2 (expert in immunology)
Remarks to the Author:

Here the authors have identified a previously unknown effect of TREM1 in atherosclerosis. They demonstrate that TREM1 is expressed in advanced human atheroma and up-regulated on peripheral and lesional myeloid cells in the *apoe*^{-/-} mice model. Furthermore, they propose that TREM1 up-regulation correlates with hyperlipidemia induced monocytosis and that it synergizes with serum factors (currently undefined) to exacerbate foam cell formation. Lastly, they report a skewed monocyte differentiation pattern and proposed CD36 up-regulation as a mechanism by which TREM1 exacerbates lesion progression. Overall the authors suggest a two phase model where TREM1 promotes monocyte production during HFCD feeding and then promotes foam cell production, resulting in larger plaques.

This is an intriguing model. Unfortunately much of the data recapitulate known functions of TREM1 such as promotion of inflammatory cytokine production, and differenced in intensity on PMN and macrophages, and up-regulation in inflammatory settings. Here we are, therefore, left with interesting new effects on hematopoiesis that are completely unexplained, and new effects on foam cell production that are also unexplained. The macrophage population stuff is largely descriptive and does not help to understand how TREM1 affects disease. Additional information here would strengthen the paper substantially.

Please mainly refer to our detailed point-by-point reply to major concerns 1-2 and minor concerns 1-7 that we provide below. In brief, although upregulated expression of TREM-1 in inflammatory settings and amplification of pro-inflammatory cytokine production are generally known functions of TREM-1, this modulation and synergistic effect of TREM-1 has not yet been described in the setting of dyslipidemia. The macrophage population data in Figure 4 admittedly are largely descriptive owing to the difficulty in isolating sufficient aortic monocyte and macrophage numbers even only for robust gene expression analysis. In particular, we have repeatedly attempted to FACS-sort macrophage subsets for qRT-PCR but even when pooling cells from 3 aortas per group of mice and using state-of-art RNA isolation techniques, CT values for the house keeping gene *Gapdh* were already at C_T 33-35 for the most abundant cell populations. With the very low expression of genes of interest ($\geq C_T$ 37) and high variability between replicate samples, we did not want to draw any conclusions from these analyses. It is noteworthy that in another study which performed gene expression profiling on total arterial macrophages "18-20 aortas were pooled per sample" (Ensan S et al., *Nat Immunol* 17,159-168 (2015)). Such analyses were beyond the scope of our facilities. However, we still believe our macrophage data of substantial interest, considering the recent identification of distinct arterial macrophage subsets in the mouse steady state aorta and the overall paucity of flow cytometry data for atherosclerotic aortas. In the *Apoe*^{-/-} model of diet-induced atherogenesis we see a strong effect of TREM-1 on atherosclerotic lesion progression which is related to aggravated monocytosis and increased monocyte/macrophage accumulation at arterial sites (Figures 1-3). Hence, it seemed highly relevant to us to provide a detailed characterization of lesional macrophage subsets and to confirm the expression of TREM-1 by monocyte/macrophage subsets by flow cytometry. While we could not further decipher the ontological origin and functions of the macrophage populations described, our descriptive data may be of value to other studies in the field, particularly, when considering the rising interest in the origins and roles of tissue-resident macrophages in health and disease.

Regarding the effects of TREM-1 on hematopoiesis and foam cell formation, we have conducted a series of additional experiments for the revised manuscript:

We show expression of TREM-1 on GMP (but not CMP or LSK cells) isolated from 16 weeks HFCD-fed *Trem1*^{+/-} *Apoe*^{-/-} mice (new Supplementary Fig. 2c) and increased expression of *Irf8* in GMP isolated from HFCD-fed *Trem1*^{+/-} *Apoe*^{-/-} mice (revised Fig. 2f). We have further FACS-sorted GMP for stimulation with anti-TREM-1 *in vitro* and obtained data which rather argue against a simple cell-autonomous effect of TREM-1 in driving skewed monocyte differentiation (new Supplementary Fig. 2d,e).

In the revised Figure 6, we have inserted new data panels (Fig. 6e-g) which demonstrate that TREM-1 also promotes *CD36* expression and foam cell formation in primary human monocytes as well as profoundly altering the expression of several genes involved in cellular cholesterol metabolism.

We are confident that with the additional data and carefully revised changes in the text the paper indeed has gained in clarity and strength. Consequently, we would like to thank Reviewer 2 for all the concise and helpful comments.

Additional Major concerns:

1- Since the histological evaluation of atherosclerotic lesions in the *apoe*^{-/-} mice model was the foundation for the subsequent findings, evaluation of the aortic sinus at 16 weeks post HFCD is critical to fully understand the mechanism by which TREM1 regulates lesion formation. This anatomical site was evaluated at an earlier time point (4 weeks post HFCD) but not in the advance stage. Moreover if as suggested monocytes and foam cell formation are promoted by TREM1, why is this effect not seen at 4 weeks?

We apologize for our inattention to this important point. In the revised Figure 1, we have now included representative pictures of the aortic sinus at 16 weeks post HFCD (revised Fig. 1e) and the calculation of the total lesion area (revised Fig. 1f). In contrast to the significant impact of TREM-1 on the extent of atherosclerotic surface area in the aorta, this effect was not revealed in the analysis of the cross-sectional lesion area at the aortic root (revised Fig. 1f). We believe that the most likely explanation for this discrepancy is in fact the mechanism by which TREM-1 influences atherogenesis. In particular, our results indicate that TREM-1 primarily affects atherosclerosis by augmenting dyslipidemia-induced monocytes (Fig. 2a). However, as it has also been shown by others (e.g. Murphy et al., J Clin Invest. 2011 Oct;121(10) dyslipidemia-associated monocytes develops in a progressive manner only after prolonged HFCD feeding. Indeed, the effect of TREM-1 on monocytes only becomes significant after 16 weeks of HFCD feeding (Fig. 2a) and TREM-1 may therefore not influence the early steps of atherogenesis. In line with this notion, our analysis of early atherogenesis did not reveal differences between *Trem1*^{+/+} *Apoe*^{-/-} and *Trem1*^{-/-} *Apoe*^{-/-} mice at 4 weeks post HFCD (previous Fig. 1e,f; new Supplementary Fig. 1a,b). Therefore, TREM-1-driven monocytes will likely enhance the progression of already established lesions, which rather translates into a lateral lesion growth (corresponding to the changes in the surface area evaluated in the aorta), but not into a significant increase in lesion thickness (cross-sectional area evaluated at the aortic root). In addition, lesions at the aortic root are usually the first to develop and represent the most advanced lesions throughout the aortic tree. As a result, these lesions might have been too advanced at 16 weeks post HFCD to still reflect the effects of TREM-1 in lesion progression, especially since lateral lesion growth cannot be evaluated at the aortic root.

We have now modified Figure 1 to comprise both the aorta (Fig. 1a-d) and aortic sinus data (revised Fig. 1e,f) for the 16 weeks post HFCD time-point. For more clarity, we have moved the 4 weeks post HFCD data (previous Fig. 1e,f) to a new Supplementary Figure 1 (Supplementary Fig. 1a,b) and refer to it in the revised Discussion: (p 12-13, lines 393-396). Importantly, we have re-phrased all previous passages stating that TREM-1 deficiency *protects* from atherosclerosis by saying that TREM-1 *attenuates* atherosclerotic disease progression. Moreover, in the Discussion section of the revised manuscript we have re-written the second paragraph to explain the seeming discrepancy between the aorta and aortic sinus data and to further clarify the proposed link between TREM-1-driven monocytes and atherosclerotic lesion progression to the reader (p 12-13, lines 384 - 405).

2- To properly characterize the *in vivo* effects of TREM1 on vascular inflammatory cells the gene expression profiling in the aorta should have been normalized to a monocyte/macrophage marker to account for macrophage lesion and infiltration differences already described. Although the authors argued they restricted the analysis to the aortic arch, where "the least differences in total lesion size was observed", a bias is clear due to up-regulation of common lesional macrophage markers (figure 3D). Furthermore, why does TREM1 does not come up in the gene profiling analysis further validating the previous qPCR data (figure 3A).

Please see also our reply to Reviewer 1 regarding the rationale of our approach of an unbiased gene expression profiling and the feasibility of assessing gene expression in purified aortic macrophages. While the data in Figure 2 indicate that the TREM-1-driven monocytes may be an underlying cause for increased monocyte infiltration and lesion progression in *Trem1*^{+/+} *Apoe*^{-/-} mice, up to Figure 3 we have not yet described that the increased extent of atherosclerosis in *Trem1*^{+/+} *Apoe*^{-/-} mice in fact associates with increased monocyte/macrophage infiltration of the aortas. In particular, TREM-1 is expressed at high levels on neutrophils which could also contribute to atherosclerotic lesion progression. (In other disease models employed in our laboratory such as colitis and colorectal cancer models, expression of *Trem1* at inflamed sites correlates with neutrophils but not monocytes/macrophages).

Hence, Figure 3 builds up on Figure 2 to demonstrate that *Trem1* is indeed expressed also at the aortic site and that the increased abundance of *Trem1* transcripts at 16 weeks HFCD primarily associates with increased expression of monocyte/macrophage-related genes at this time-point. We appreciate the Reviewer's point that these data cannot be considered particularly novel or surprising. However, we believe that the gene expression data in Figure 3 are required to corroborate what is proposed in Figure 2: (i) The effects of TREM-1 on atherosclerosis are only evident after prolonged HFCD (but not chow) feeding, and (ii) aggravated peripheral blood monocytes in HFCD-fed *Trem1*^{+/+} *Apoe*^{-/-} mice translates into increased monocyte/macrophage infiltration of aortas.

As requested by Reviewer 2, we have normalized the gene expression data of the previous Fig. 3d to a set of monocyte/macrophage markers (Figure R2.1 for Reviewer 2). With this approach, distinct differences in gene expression between *Trem1*^{+/+} *Apoe*^{-/-} and *Trem1*^{-/-} *Apoe*^{-/-} mice are no longer apparent (Figure R2.1 for Reviewer 2). Although this may seem discordant with the clear effects of anti-TREM-1 stimulation on human monocytes *in vitro* (i.e. upregulated expression of pro-inflammatory cytokines and lipid metabolism-associated genes; Figure 5 and revised Figure 6), it has to be considered that aortic wall-infiltrating monocytes/macrophages represent a heterogeneous population (Figure 4) and that TREM-1-mediated signaling could distinctly affect gene expression only in a subset of these cells (e.g. newly infiltrating monocytes). Moreover, also the expression of

monocyte/macrophage markers could be subject to inflammation-induced changes.

Finally, *Trem1* did come up in the Nanostring profiling analysis (first row in heat map of previous Fig. 3c). We have included the raw Nanostring data for *Trem1* and *Trem2* as well as qRT-PCR data for *Trem1* and *Trem14* (which were not included in the panel) in a new Supplementary Figure 3 (Supplementary Fig. 3a,b) which now clearly shows expression levels for the individual mice. Moreover, in order to simplify the previous Figure 3, we have moved the panels showing individual expression of candidate genes (previous Fig. 3d) also to the new Supplementary Figure 3 (Supplementary Fig. 3c). The revised Figure 3 now instead shows heat-maps of the selected gene categories with mean log-fold changes in gene expression in 16 weeks HFCD-fed *Trem1*^{+/+} *Apoe*^{-/-} versus *Trem1*^{+/+} *Apoe*^{-/-} mice (revised Fig. 3c). We believe that with this approach the main take-home messages of Figure 3 (i.e. expression of *Trem1* at arterial sites, requirement for HFCD to reveal the effects of TREM-1, association of *Trem1* with increased monocyte/macrophage accumulation) are better conveyed to the reader without placing too much emphasis on the admittedly rather confirmatory data on TREM-1-mediated pro-inflammatory effects.

Minor concerns:

1- The statistical analysis seems unclear. In many cases, points with wildly overlapping error bars are denoted as significantly different. It does not seem that that could be the case. See 2a and 5a

We agree that the error bars in Fig. 2a appear rather disadvantageous. However, it has to be considered that Fig. 2a shows pooled *ex vivo* data from 3 independent experiments which were running over 16 weeks. In fact, the greatest variability is observed for the 16 weeks time-points, not for the other time-points. Moreover, error bars indicate the SD, not the SEM. Despite the overlapping error bars, data are significantly different where indicated - owing to the fact that a total of n=13-15 mice was analyzed.

While we have previously attempted to plot the data in Fig. 2a in different ways (e.g. with symbols showing values for individual mice), we still believe that the current presentation of the data most clearly and legibly illustrates the progressive increase in monocytes over neutrophils in *Trem1*^{+/+} *Apoe*^{-/-} versus *Trem1*^{-/-} *Apoe*^{-/-} mice.

Similar to the long-term *in vivo* experiments in mice, variability has to be taken into account for experiments involving human blood donors. Here (Fig. 5c and revised Fig. 6g), we have aimed for maximal transparency by plotting absolute values and showing individual data for the different donors. Importantly, the heterogeneous cytokine response of primary monocytes to anti-TREM-1 stimulation does not represent an *in vitro* artefact but relates to intrinsic differences in the extent of TREM-1 surface expression and responsiveness to TREM-1 stimulation of human individuals (unpublished data and Saurer et al., J Crohn's Colitis. 2012 Oct;6(9)).

For both Figure 2 and Figure 5, the number of mice / human individuals analyzed and the statistical testing employed have been described in the accompanying Figure legends. However, to make it clearer to the reader that Figure 5 comprises 6-7 independent experiments involving human monocytes from different blood donors, we have slightly modified the legend to Figure 5: p 32, lines 1013-14 and 1020-21:

2- Since the authors suggest that TREM1 signaling induces monocytic skewing of hematopoiesis in the bone marrow, this would suggest that these progenitors express TREM1? Is this the case?

Yes, GMP, but not CMP or LSK cells, clearly express surface TREM-1. This has been described by us in our previous characterization of *Trem1*^{-/-} mice (Weber B et al., PLoS Pathog. 2014 Jan;10(1)) and was also referred to in the Discussion section of the previous manuscript. However, owing to the helpful suggestion of the reviewer, we now realize that this information is very central and should be directly provided to the reader. Thus, in the new Supplementary Figure 2 of the revised manuscript, we show the staining for surface TREM-1 on LSK cells, CMP and GMP isolated from the bone marrow of 16 weeks HFCD-fed versus chow-fed *Trem1*^{+/+} *Apoe*^{-/-} mice along with the gating strategies for these cells (Supplementary Fig. 2b,c). Importantly, these data confirm our previous findings on the distinct expression of TREM-1 by GMP (but not CMP or LSK cells) and also demonstrate that HFCD feeding (compared to chow) does not further upregulate expression of TREM-1 on progenitors (Supplementary Fig. 2c).

Since the expression of TREM-1 by GMP nonetheless suggests that TREM-1-mediated signaling could directly induce monocytic skewing in a cell-autonomous manner, in an additional set of experiments for the revised manuscript, we have sorted GMP from 16 weeks HFCD-fed mice for *in vitro* activation by plate-bound anti-TREM-1 mAb in the presence or absence of HFCD serum. Differentiation of monocytic versus granulocytic cells was determined after 72 h of culture by flow cytometry. As shown in the new Supplementary Figure 2, direct activation of GMP *via* TREM-1 had no appreciable impact on the differentiation of GMP into monocytes or granulocytes irrespective of the presence or absence of HFCD serum in the medium (Supplementary Fig. 2d,e). While these results *may* indicate that the main effect of TREM-1 on monocytic skewing could be indirect and/or upstream of GMP, this was difficult to conclusively address within the granted revision time. We have analyzed sera as well as supernatants from flushed bones for cytokines using multiplex assays but values for cytokines were generally close to the detection limit and we did not find major differences for cytokines such as M-CSF or G-CSF which could have explained the effect of TREM-1 on myelopoietic processes (data not shown). We are convinced that the role of TREM-1 in myelopoiesis merits further attention. However, as such studies have to be crucially designed to incorporate the complex effects of dyslipidemia *in vivo* and also need to run over extended time-periods to unmask the effects of TREM-1 (revised Fig. 2a), we will follow this as a separate project.

Still, in the revised manuscript we are able to show that GMP from HFCD-fed *Trem1*^{+/+} *Apoe*^{-/-} mice tend to

express increased levels of the transcription factor *Irf8* (revised Fig. 2f), which has been described as a key regulator of monocyte lineage specification (reviewed in Yáñez, A et al., *Curr Opin Hematol.* 2016 Jan;23(1). Moreover, in the text of the revised manuscript we have inserted the following additions/changes: Results: p6, lines 169 - 177; Discussion: p13, lines 410-429.

3- TREM1^{-/-} should be shown in 2F

The MFI values for *Trem1^{-/-} Apoe^{-/-}* mice have now been included in the revised Fig. 2h (previous Fig. 2f).

4- The gene expression data does not seem to offer anything to the paper. In fact, CD36 is shown to be not different yet this is pursued in the paper

Please also see our detailed reply to the major concern 2. In brief, we believe that the data in Figure 3 is required to substantiate that the increased peripheral blood monocytoysis in *Trem1^{+/+} Apoe^{-/-}* mice described in Figure 2 in fact translates into increased aortic infiltration with monocytes/macrophages and that increased abundance of *Trem1* transcripts at 16 weeks post HFCD (but not chow) associates with higher expression levels of monocyte/macrophage-related genes. Importantly, also in advanced human atheromas the upregulated expression of *TREM1* is paralleled by an increased expression of *CD68* but not *CD15* (Fig. 8). This is in fact a significant finding when considering that in peripheral blood myeloid cells TREM-1 is highly expressed on neutrophils (revised Fig. 2g).

Moreover, while TREM-1-mediated signaling clearly affects the expression of *CD36* and other cholesterol metabolism-related genes in human monocytes *in vitro* (revised Fig. 6g), such effects may easily be masked when analyzing entire tissues where even monocyte/macrophages represent a heterogenous population. The impact of TREM-1 on foam cell formation, expression of *CD36* and other genes involved in lipid metabolism was evaluated as a possibility using additional *in vitro* experiments, because the gene expression profiling *ex vivo* did not indicate a single/unequivocal gene/receptor/pathway accounting for TREM-1-mediated lesion progression. Considering that innate immune signaling pathways are known to interfere with cellular cholesterol homeostasis, it seemed a reasonable proceeding to also look at the direct impact of TREM-1 on foam cell formation.

As shown in the revised Figure 6, the effects of TREM-1-mediated signaling on foam cell formation, expression of *CD36* and other genes involved in cholesterol metabolism are also observed in primary human monocytes (revised Fig. 6e-g), lending further support to this unanticipated novel function of TREM-1.

Nonetheless, we fully appreciate that the previous Figure 3 was too bulky and overrated. Hence, we have moved all the individual gene expression data panels from the previous Fig. 3d to a new Supplementary Fig. 3c. The revised Figure 3 has been reduced/modified to only show: (Fig. 3a) Upregulated expression of *Trem1*, *Cd68* and *Ly6g*, (Fig. 3b) PCA plots, and (Fig. 3c) heat-maps of the selected gene categories with mean log-fold changes in gene expression in 16 weeks HFCD-fed *Trem1^{+/+} Apoe^{-/-}* versus *Trem1^{-/-} Apoe^{-/-}* mice.

5- Soluble TREM1 is not shown at all. Does HFCD yield sTREM1? Do patients with advanced disease have high sTREM1 levels?

In the revised Figure 2i, we now also show data for serum sTREM-1 in chow-fed versus HFCD-fed *Trem1^{+/+} Apoe^{-/-}* mice. Intriguingly, compared to 16 weeks chow-fed mice, levels of sTREM-1 are distinctly elevated after 4 weeks of HFCD. These data support the notion that factors contained in dyslipidemic serum can impact on the surface expression (Fig. 2g,h) and hence also the shedding of TREM-1 by peripheral blood myeloid cells. Unfortunately, we were not able to access sera from patients with advanced disease as primary inclusion sera of individuals comprised in the Angiolaus Study were no longer available. However, studies by others indicate that serum sTREM-1 may indeed also be upregulated in patients with coronary artery disease (Hermus L et al., *Clin Biochem.* 2011 Nov;44(16)).

6- Other members of the TREM family have been or are likely involved in atherosclerosis. Specifically, Trem14 is known to be involved, Trem2 has been implicated and Trem11 is a platelet gene. Does the elimination of TREM1 affect the expression of any of these other genes?

In the new supplementary Figure 3a,b we now show panels with the expression levels of *Trem1*, *Trem2* (Nanostring and qRT-PCR), *Trem11* and *Trem14* (qRT-PCR, since not included in the original Nanostring panel). *Trem2* is also expressed at higher levels in *Trem1^{+/+} Apoe^{-/-}* compared to *Trem1^{-/-} Apoe^{-/-}* mice while there is no significant difference in expression levels of *Trem11* and *Trem14*. The increased expression of *Trem2* is most likely attributed to the overall increased infiltration with monocytes/macrophages in *Trem1^{+/+} Apoe^{-/-}* mice. Although the dogma is that TREM-1 counteracts TREM-2 (and *vice versa*), in our comprehensive characterization of *Trem1^{-/-}* mice, we have not found evidence for increased TREM-2-mediated functions (Weber B et al., *PLoS Pathog.* 2014 Jan;10(1). In fact, TREM-2 has been postulated to be generally induced on tissue-infiltrating monocytes as they differentiate into macrophages (our own observations and Turnbull IR et al., *J Immunol* 2006 Sep 15;177(6)). While *TREML4* evidently has been implicated in the formation of calcified atheromatous plaques in humans, expression levels of *Trem14* were comparable in *Trem1^{+/+} Apoe^{-/-}* versus *Trem1^{-/-} Apoe^{-/-}* mice. Hence, at least in our model, *Trem14* expression on monocytes/macrophages does not appear to play a major part.

7- Foam cell formation and the regulation of this process should be assessed in primary monocytes from humans.

In the revised Figure 6, we now provide additional sets of *in vitro* data involving human primary monocytes. These data demonstrate that TREM-1 (but not LPS) promotes foam cell formation also in CD14^{hi} monocytes (revised

Fig. 6e,f), the equivalent subset of murine Ly6C^{hi} monocytes which are acknowledged precursors of foam cells *in vivo*. Importantly, TREM-1 promotes upregulated expression of *CD36* also in human primary monocytes. The increase in *CD36* is paralleled by augmented expression of *PPARG*, lending support to the notion that *PPARG* is a TREM-1-regulated gene and *PPARG* in turn is a key regulator of *CD36*. However, our additional data also show that the effect of TREM-1 is not restricted to *CD36* but that TREM-1 affects the expression of several genes implicated in cellular cholesterol metabolism, including other scavenger receptors (*MSR1*) cholesterol efflux-related transporters (*ABCA1*, *ABCG1*) and intracellular cholesterol transport proteins (*NPC1*, *NPC2*, *LIPA*, *STARD4*). This novel information is shown in the revised Fig. 6g.

Reviewer #3 (expert in atherosclerosis)
Remarks to the Author:

In the manuscript "TREM-1 links dyslipidemia to inflammation and lipid deposition in atherosclerosis" Zysset et al. profile development of atherosclerosis in *Apoe*^{-/-} *Trem-1*^{-/-} mice. They show that the DKO have smaller lesions without effects on lipids, and less monocytois. Many inflammatory genes are attenuated in the aortas of the DKOs, as demonstrated by gene expression profiling. TREM-1 is augmented on circulating monocytes after high fat diet and is expressed somewhat on monocytes and neutrophils in the aorta. Mechanistically, the authors suggest that oxLDL upregulates TREM-1 and TREM-1 increases cytokine production. They also suggest that TREM-1 augments *CD36* and foam cell formation. Finally, the authors show that TREM-1 is expressed in human lesions. Overall, this is a good paper.

1. The authors should show their gating strategy for the various HSPC as shown in Figure 2.

A gating strategy for HSPC was also requested by Reviewer 1. Gating strategies for both peripheral blood myeloid cell subsets as well as their upstream bone-marrow precursor cells are now included in a new Supplementary Figure 2 (Supplementary Fig. 2a,b).

2. TREM-1 is only mildly augmented in the various cell subsets, as shown by flow cytometry in Figure 4. It would be reassuring to see TREM-1 expression on sorted cells by either PCR (for message) or on Western blots (for protein).

We appreciate the Reviewer's concern regarding these data. However, we have used matched isotype control stained cells for the histogram overlays (Fig. 4c). Moreover, surface expression of TREM-1 on myeloid cells other than neutrophils generally is low (Fig. 2g). In this respect, or comparing the expression levels of TREM-1 on peripheral blood versus aortic neutrophils, we consider the expression of TREM-1 on aortic monocyte and macrophage subsets quite substantial. In other words, the lower expression of TREM-1 on aortic compared to peripheral blood neutrophils indicates that either TREM-1 was partially shed on aortic neutrophils or that TREM-1 was somewhat sensitive to the aortic enzymatic digestion protocol. Since TREM-1 still is expressed at high levels on neutrophils isolated from inflamed colons (Weber B et al., *PLoS Pathog.* 10 (2014) and unpublished data), we believe that the lower expression of surface TREM-1 on aortic subsets is not just due to inflammation-induced shedding but rather relates to the harsher aortic digestion protocol.

While we agree that a confirmation of TREM-1 expression by qRT-PCR or even Western blots would have been highly desirable, our repeated attempts to FACS-sort sufficient cells for robust RNA isolation have not been successful. Specifically, even when pooling cells from 3 aortas and focusing on the most abundant subsets (MHC class II⁺ and MHC class II⁻ macrophages), CT values for the house keeping gene *Gapdh* were already at C_T 33-35. With the very low expression of genes of interest ($\geq C_T$ 37) and high variability between replicate samples, we did not want to draw any conclusions from these analyses.

However, to make the potential impact of the aortic digestion protocol on TREM-1 surface expression clearer to the reader we have added the following sentence to the Results section: p9, lines 258 - 259.

Figure R2.1 for Reviewer 2

Nanostring-based gene expression data after normalization to combined macrophage markers.

Expression levels of *Cd14*, *Cd68*, *Cd74*, *Csf1r*, *Cx3cr1*, *Emr1*, *Fcgr1* and *Itgam* were used to calculate a mean combined macrophage marker expression level. For each sample, the expression of a candidate gene was calculated relative to the mean macrophage marker expression level. Column graphs show mean mRNA expression levels of genes of interest relative to macrophage-associated gene expression for each group of mice.

REVIEWERS' COMMENTS:

Reviewer #1 (Remarks to the Author):

The manuscript is improved; however, the authors only partially responded to the concerns of the reviewer. Much of the data remain descriptive; not helped by the fact that the gene expression data on 'macrophages' is from whole aorta. The Facs gating in Figure 4 is incorrect.

Reviewer #2 (Remarks to the Author):

The revised manuscript by Zysset et al. is substantially improved by the additional data and the clarifications in the text. In particular, the changes to the histology figures, the increase in information regarding the hematopoiesis, and the human foam cell information add strength to the paper. Given the caveats noted by the authors in response to my concern regarding the gene expression data it is understandable that the conclusions are limited. However, I feel that a note within the text clearly stating that correction for myeloid cell infiltrate negates the qualitative differences reported is important to make it clear that this global approach does not reveal Trem1-mediated differences in these populations rather the expression differences are reflective of greater monocyte infiltration. The confounding issues suggested in the response to reviewers could be included but its important to convey that this observation does not recapitulate the in vitro findings and does not suggest in situ differences in these genes on a per cell basis.

Otherwise the authors have sufficiently addressed my concerns.

Reviewer #3 (Remarks to the Author):

The authors addressed my few concerns.

Point-by-point reply to Reviewers' comments on revised manuscript NCOMMS-16-05799A "TREM-1 links dyslipidemia to inflammation and lipid deposition in atherosclerosis"

The authors would like to thank all the Reviewers for taking their time to study our revised manuscript and our previous point-by-point reply. In the current point-by-point-reply, the original Reviewers' comments have again been copied in blue font color while our answers are marked in black font color. Changes to the text of the revised manuscript have been applied with the "track changes" function, as requested by the Editorial Board.

Reviewer #1 (Remarks to the Author):

The manuscript is improved; however, the authors only partially responded to the concerns of the reviewer. Much of the data remain descriptive; not helped by the fact that the gene expression data on 'macrophages' is from whole aorta. The FACS gating in Figure 4 is incorrect.

In response to the comment on the "incorrect" FACS gating raised by Reviewer 1, but also in agreement with the additional comments and suggestions made by Reviewer 3 (to Vesna Todorovic) on how to improve the aortic FACS data, in the re-revised Supplementary Figure 5b (previous Supplementary Figure 4) we now show surface expression of Ly6G to illustrate that the gated Ly6C^{int} MHC class⁻ cells in fact correspond to Ly6G⁺ neutrophils while Ly6G is not expressed by any of the other gated myeloid cell subsets. In the Supplementary Figure 5c, we further show surface expression of the macrophage marker F4/80 (stainings for F4/80 were included in some, but not all experiments, since we prefer the CD64 marker for the identification of macrophages) on the gated myeloid subsets. Importantly, the F4/80 staining confirms that Ly6C⁻ MHC class II⁺ and MHC class II⁻ cells indeed are *bona fide* (differentiated) macrophages and that F4/80 is only weakly expressed on Ly6C^{hi} monocytes and Ly6C^{int} (incompletely differentiated) macrophages (Supplementary Fig. 5c).

While we are aware that many studies employ a pre-gating of F4/80⁺ cells to describe macrophage populations, we did deliberately not employ such an approach as this would not allow to visualize Ly6C^{hi} monocytes and Ly6C^{int} macrophages which are F4/80^{lo} (Supplementary Fig. 5c). Instead we used a gating strategy based on segregation of cells according to their Ly6C and MHC class II expression that was adapted from the monocyte waterfall concept of intestinal macrophage differentiation (Tamoutounour et al., Eur. J. Immunol. 2012; Bain C.C. et al., Mucosal Immunol. 2013) and also allows to visualize Ly6C^{hi} monocytes and their Ly6C^{int} descendants.

We realize that the rationale for our gating strategy may have been difficult to comprehend in the previous version of the manuscript. Hence, to further clarify our FACS gating approach to the reader, we have inserted the following additional text in the Results section:

Results, p8:

Within the remaining CD11b⁺ population we distinguished five myeloid cell subsets based on their expression of Ly6C and MHC class II: 1. Neutrophils (Ly6C^{int}, MHCII⁻) which were also Ly6G⁺ (Supplementary Fig. 5b), 2. Ly6C^{hi} monocytes (Ly6C^{hi}, MHCII⁻) 3. Ly6C^{int} macrophages (Ly6C^{int} MHCII⁺) 4. MHCII⁺ macrophages (Ly6C^{lo} MHCII⁺) and 5. MHCII⁻ macrophages (Ly6C^{lo}, MHCII⁻) (Fig. 4a,b). The identity of macrophages was confirmed by staining for CD64 (or F4/80 in some experiments), which was largely absent on neutrophils and expressed at higher levels in macrophages as compared to Ly6C^{hi} monocytes (Fig. 4b). Importantly, the segregation of myeloid subsets according to their Ly6C and MHC class II expression was adapted from the monocyte waterfall concept of macrophage differentiation^{45,46}, which also considers newly recruited Ly6C^{hi} monocytes and Ly6C^{int} intermediate macrophage differentiation stages. In contrast, a general pre-gating of F4/80⁺ cells *a priori* excludes these subsets since Ly6C^{hi} monocytes and Ly6C^{int} cells unlike MHCII⁺ and MHCII⁻ macrophages are F4/80^{lo} (Supplementary Fig. 5c).

Reviewer #2 (Remarks to the Author):

The revised manuscript by Zysset et al. is substantially improved by the additional data and the clarifications in the text. In particular, the changes to the histology figures, the increase in information regarding the hematopoiesis, and the human foam cell information add strength to the paper. Given the caveats noted by the authors in response to my concern regarding the gene expression data it is understandable that the conclusions are limited. However, I feel that a note within the text clearly stating that correction for myeloid cell infiltrate negates the qualitative differences reported is important to make it clear that this global approach does not reveal Trem1-mediated differences in these populations rather the expression differences are reflective of greater monocyte infiltration. The confounding issues suggested in the response to reviewers could be included but its important to convey that this observation does not recapitulate the in vitro findings and does not suggest in situ differences in these genes on a per cell basis.

Otherwise the authors have sufficiently addressed my concerns.

We have inserted additional text in the Results section in order to clearly state that correction for the myeloid cell

infiltrate negates differences in gene expression between the two groups of mice. The text now reads as follows:

Results, p7:

Accordingly, also the expression of several pro-inflammatory cytokines, oxidases and genes involved in foam cell formation was augmented (Fig. 3c and Supplementary Fig. 3c). Normalization of these differentially expressed genes to monocyte/macrophage related markers did not demonstrate significant differences between the two groups of mice (Supplementary Fig. 4). Hence, the distinct transcriptional profile of whole aortic tissue from HFCD-fed *Trem1*^{+/+} *ApoE*^{-/-} mice was mainly reflective of increased monocyte/macrophage accumulation rather than altered gene expression patterns on a per cell level.

We have further included the monocyte/macrophage marker-normalized gene expression data (as shown previously in a new Supplementary Fig. 4 (as shown previously to Reviewer 2).

Finally, we have also introduced a small text modification in the Discussion section in order to be more careful with our previous statement that TREM-1 promotes atherosclerotic lesion progression locally:

Discussion, p16:

In support of our hypothesis that TREM-1 may promote atherosclerotic lesion progression locally, we found highly upregulated expression of mRNA for TREM-1 also in human aortic tissue specimens displaying severe atherosclerosis.

Reviewer #3 (Remarks to the Author):

The authors addressed my few concerns.

Point-by-point reply to the comment of Reviewer 3 on the second revision of the manuscript NCOMMS-16-05799B

The authors responded adequately to the additional points. The new data shown in Supplementary Fig 5b and c are as I had expected. However, the argument to use the waterfall strategy instead of F4/80 is not really convincing. One can use CD11b vs F4/80 as a first gate to distinguish monocytes from macrophages and then gate on F4/80 neg CD11b+ cells for a further gate on Ly-6C vs Ly-6G (or CD115) (Theurl Nat Med 2016). In any case, I'm satisfied with the gating scheme the authors provide here.

We fully appreciate that the gating strategy proposed by Reviewer 3 is more widely employed and may appear more rational to the majority of readers. We have tried to separate populations according to CD11b vs. F4/80 / CD64 and further sub-gate the F4/80⁻ or CD64⁻ CD11b⁺ cells. However, while this resulted in two distinct MHCII⁺ and MHCII⁻ populations among the F4/80⁺ or CD64⁺ CD11b⁺ cells, the F480⁻ CD64⁻ CD11b⁺ population was highly heterogeneous, containing not only the desired Ly6G⁺ neutrophils, Ly6C^{hi} and Ly6C^{int/lo} MHCII⁻ monocytes but still also Ly6C⁻ MHCII⁺ cells, representing "contaminant" macrophages.

A more stringent/narrow setting of gates to pull the F4/80⁺ / CD64⁺ vs. CD11b⁺ cells apart, did get rid of the "contaminant" MHCII⁺ macrophages in the F4/80⁻ / CD64⁻ CD11b⁺ gate, but also resulted in the disappearance of Ly6C^{int} F480^{lo/-} MHCII⁺ macrophages (which we regarded as monocytes in the process of differentiation to mature macrophages). These cells, however, we considered as major population of interest based on their pronounced expression of TREM-1 (Fig. 4c, d).

Thus, we still prefer our waterfall gating strategy (Ly6C vs. MHCII) which displays all monocyte / macrophage differentiation stages. It is of note, that this strategy is not only employed for gating of intestinal monocyte/macrophages but has also been successfully applied for the identification/discrimination of DCs, monocytes and macrophages in the mouse skin (Tamoutounour S et al., Immunity 2013).